# Implicit Search via Discrete Diffusion: A Study on Chess

**Jiacheng Ye**[1], **Zhenyu Wu**[2], **Jiahui Gao**[3], **Zhiyong Wu**[4]
**Xin Jiang**[3], **Zhenguo Li**[3], **Lingpeng Kong**[1]
[1] The University of Hong Kong    [2] Shanghai Jiaotong University
[3] Huawei Noah's Ark Lab    [4] Shanghai AI Laboratory
carsonye@connect.hku.hk

## Abstract

In the post-AlphaGo era, there has been a renewed interest in search techniques such as Monte Carlo Tree Search (MCTS), particularly in their application to Large Language Models (LLMs). This renewed attention is driven by the recognition that current next-token prediction models often lack the ability for long-term planning. Is it possible to instill search-like abilities within the models to enhance their planning abilities without relying on explicit search? We propose DIFFUSEARCH, a model that does *implicit search* by looking into the future world via discrete diffusion modeling. We instantiate DIFFUSEARCH on a classical board game, Chess, where explicit search is known to be essential. Through extensive controlled experiments, we show DIFFUSEARCH outperforms both the searchless and explicit search-enhanced policies. Specifically, DIFFUSEARCH outperforms the one-step policy by 19.2% and the MCTS-enhanced policy by 14% on action accuracy. Furthermore, DIFFUSEARCH demonstrates a notable 30% enhancement in puzzle-solving abilities compared to explicit search-based policies, along with a significant 540 Elo increase in game-playing strength assessment. These results indicate that implicit search via discrete diffusion is a viable alternative to explicit search over a one-step policy. All codes are publicly available at https://github.com/HKUNLP/DiffuSearch.

## 1 Introduction

Search is central to problem-solving in AI (Russell & Norvig, 2010). One of the most notable examples is IBM's Deep Blue (Campbell et al., 2002), which performs extensive search over a large space through a strong search algorithm (alpha-beta pruning; Knuth & Moore 1975), defeated the world chess champion Garry Kasparov in 1997. Search has also been utilized in neural networks. A noteworthy advancement in this progression is exemplified by AlphaGo (Silver et al., 2016) and its successors (Silver et al., 2017b;a), where the policy is guided by an extra value network through Monte Carlo Tree Search (MCTS; Coulom 2006; Browne et al. 2012). By explicitly searching into the future, the decision to be taken can be iteratively refined (Silver et al., 2016; 2017b;a; Schrittwieser et al., 2020).

Recent research on Large Language Models (LLMs) demonstrates the utilization of a similar framework. Although scale-up, the autoregressive one-step policy addresses only a portion of the problems and relies on explicit search on complex tasks (Hao et al., 2023; Yao et al., 2024; Zhao et al., 2024; Trinh et al., 2024, *inter alia*), highlighting their inherent limitations in long-term planning (Valmeekam et al., 2022; Bubeck et al., 2023; Bachmann & Nagarajan, 2024). This explicit search-demanding approach, however, is not quite satisfactory, as the repeated invocation of the value model can result in an accumulation of errors if the value model is inaccurate and increased inference costs for long-horizon rollouts (Yao et al., 2024). Given the essence of explicit search (e.g., MCTS) over one-step policies lies in iteratively looking into the future and leveraging the future to enhance the next token (or action) prediction, our research question is:

*Can the policy model predict and utilize the future by itself to improve the next token (or action) prediction without relying on explicit search during inference?*

This paper explores the potential transition from utilizing an explicit search algorithm (e.g., MCTS) over the one-step policy to implicitly searching over future representations by teaching the policy to predict and utilize the future. Firstly, to reduce the difficulty of future prediction, we take inspiration from diffusion models (Sohl-Dickstein et al., 2015; Ho et al., 2020), which perform a multi-step generative process for sample generation. Secondly, to iterative refine the current policy prediction based on future information, we directly rely on the internal bidirectional self-attention mechanism (Vaswani et al., 2017) and the multi-step diffusion generative process. Finally, we represent the future to be learned and predicted with the multi-step interaction information between policy and the world (e.g., states and actions), such that the generation of the future shares similar spirits as implicit searching in the future world. We name our approach as DIFFUSEARCH, a method that looks into the future world via diffusion modeling without any explicit search during inference. Alternatively, DIFFUSEARCH can be seen as containing a world model that predicts the future. However, rather than having a separate world model simulating the environment's transition dynamics and another policy performing action prediction through interaction with the world model using planning algorithms such as value iteration (Puterman, 2014) or MCTS (Schrittwieser et al., 2020), DIFFUSEARCH internalizes the world model directly within the policy without intermediate components. We show the comparison between explicit search via MCTS and implicit search via discrete diffusion in Figure 1.

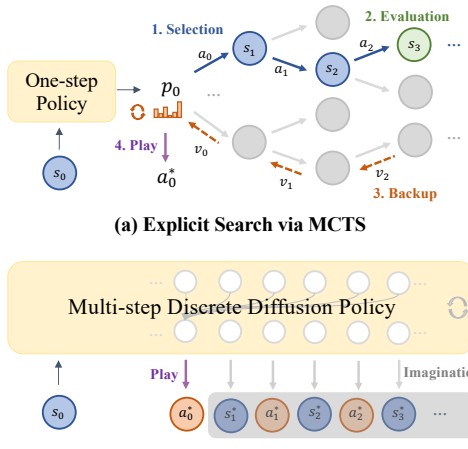

(a) Explicit Search via MCTS

(b) Implicit Search via Discrete Diffusion

Figure 1: Comparison between explicit search via MCTS and implicit search via discrete diffusion. MCTS explicitly performs action selection, state evaluation, and value backup in an iterative manner before determining the next action to take (as detailed in Appendix A), while discrete diffusion implicitly gathers future information during future imagination to improve the next action.

We take a specific focus on the chess-playing task, where explicit search is known to be essential (Campbell et al., 2002; Silver et al., 2017a). The ideas and techniques learned in this controlled task may eventually be useful in natural-language settings as well. We conduct extensive experiments and take a deep look into various paradigms to represent and learn the future. When measured by action accuracy, DIFFUSEARCH outperforms the one-step policy (Ruoss et al., 2024) by 19.2%, and MCTS-enhanced policy by 14%. DIFFUSEARCH demonstrates a 30% increase in puzzle-solving capabilities in comparison to the MCTS-enhanced policy. Furthermore, it attains a higher level of game-playing proficiency, as evidenced by a 540 more Elo rating, which showcases the potential of substituting one-step policy with explicit search with a learned discrete diffusion model that looks into the future world by itself.

Our contributions include: 1) we propose DIFFUSEARCH to foresee and utilize future information via diffusion modeling as an alternative to explicit search via designed search algorithms (§3); 2) we instantiate DIFFUSEARCH for chess-playing and demonstrate its superior performance compared to both the one-step policy and the MCTS-powered policy in a rigorous evaluation, such as solving over 30% more puzzles and 540 Elo playing strength in the tournament (§4.2); 3) we provide a detailed analysis of the design considerations for future representation and diffusion modeling (§4.3), as well as unveiling the working mechanism and appealing advantage compared to MCTS-based policy regarding effectiveness and efficiency (§4.4). These findings demonstrate the possibility of moving from the one-step policy with explicit search algorithms to the future world-aware multi-step diffusion policy with implicit search ability.

## 2 PRELIMINARIES

This section introduces key concepts and notations in the chess-playing problem and diffusion modeling.

**Problem Setting**    Chess, along with other games of perfect information like checkers, othello, backgammon, and Go, fits the framework of alternating Markov games (Littman, 1994). In chess, there exists a state space $\mathcal{S}$, an action space $\mathcal{A}$, a state transition function $f(s, a)$ that determines the subsequent state after taking action $a$ in state $s$, and two reward functions $r^0(s)$ and $r^1(s)$ representing the two players' reward in state $s$ (rewards being zero except at the final time-step). The outcome of the game $o_i = \pm 1$ is the terminal reward at the end of the game from the perspective of the current player at time-step $i$. Chess is also a zero-sum game which indicates $r^0(s) = -r^1(s)$. A policy $p(a|s)$ is a probability distribution over actions space $\mathcal{A}$. A value function $v^p(s)$ represents the expected outcome when all actions for both players adhere to policy $p$, denoted as $v^p(s) = \mathbb{E}[o_i|s_i = s, a_{i...I} \sim p]$. The goal is to build a policy that, when actions are taken based on it, results in the highest possible final outcome.

**Discrete Diffusion Modeling**    Discrete diffusion models (Sohl-Dickstein et al., 2015; Hoogeboom et al., 2021; Austin et al., 2021) are a class of latent variable models characterized by a forward and a backward Markov process. Suppose $\boldsymbol{x}_0 \sim q(\boldsymbol{x}_0)$ is a discrete random variable with $K$ possible categories and represented as a one-hot vector. The forward process $q(\boldsymbol{x}_{1:T}|\boldsymbol{x}_0) = \prod_{t=1}^{T} q(\boldsymbol{x}_t|\boldsymbol{x}_{t-1})$ corrupts the original data $\boldsymbol{x}_0$ into a sequence of increasingly noisy latent variables $\boldsymbol{x}_{1:T} \coloneqq \boldsymbol{x}_1, \ldots, \boldsymbol{x}_T$. The learned backward process $p_{\boldsymbol{\theta}}(\boldsymbol{x}_{0:T}) = p(\boldsymbol{x}_T) \prod_{t=1}^{T} p_{\boldsymbol{\theta}}(\boldsymbol{x}_{t-1}|\boldsymbol{x}_t)$ gradually denoises the latent variables to the data distribution. In order to optimize the generative model $p_{\boldsymbol{\theta}}(\boldsymbol{x}_0)$ to fit the data distribution $q(\boldsymbol{x}_0)$, we typically optimize a variational upper bound on the negative log-likelihood due to the intractable marginalization:

$$L_{\text{vb}} = \mathbb{E}_{q(\boldsymbol{x}_0)} \left[ \underbrace{D_{\text{KL}}[q(\boldsymbol{x}_T|\boldsymbol{x}_0)||p(\boldsymbol{x}_T)]}_{L_T} + \sum_{t=2}^{T} \underbrace{\mathbb{E}_{q(\boldsymbol{x}_t|\boldsymbol{x}_0)}\left[D_{\text{KL}}[q(\boldsymbol{x}_{t-1}|\boldsymbol{x}_t,\boldsymbol{x}_0)||p_{\boldsymbol{\theta}}(\boldsymbol{x}_{t-1}|\boldsymbol{x}_t)]\right]}_{L_{t-1}} \right.$$
$$\left. \underbrace{- \mathbb{E}_{q(\boldsymbol{x}_1|\boldsymbol{x}_0)}[\log p_{\boldsymbol{\theta}}(\boldsymbol{x}_0|\boldsymbol{x}_1)]}_{L_0} \right], \tag{1}$$

where $L_T$ is a constant when a fixed prior $p(\boldsymbol{x}_T)$ is employed. In discrete diffusion, both the forward and backward distribution are defined as categorical distribution, e.g., $q(\boldsymbol{x}_t|\boldsymbol{x}_{t-1}) = \text{Cat}(\boldsymbol{x}_t; \boldsymbol{p} = \boldsymbol{Q}_t^\top \boldsymbol{x}_{t-1})$ and $p_{\boldsymbol{\theta}}(\boldsymbol{x}_{t-1}|\boldsymbol{x}_t) = q(\boldsymbol{x}_{t-1}|\boldsymbol{x}_t, f(\boldsymbol{x}_t; \boldsymbol{\theta}))$ (Hoogeboom et al., 2021), where $\boldsymbol{Q}_t$ is a pre-defined transition matrix of size $K \times K$. Therefore, the forward process posterior $q(\boldsymbol{x}_{t-1}|\boldsymbol{x}_t, \boldsymbol{x}_0)$ and each KL term can be calculated analytically. We provide more details about discrete diffusion in Appendix B.1.

## 3    METHODOLOGY

In this section, we introduce DIFFUSEARCH, an approach that looks into the future world via discrete diffusion modeling without any explicit search at inference time, with a focus on the chess-playing task.

### 3.1    MODELING

In order to endow the model with the capability to predict and utilize the future, we consider training the model in a supervised way following (Ruoss et al., 2024), leaving self-play training from scratch (Silver et al., 2017b) for future work. We provide the current state $s_i$ as the history representation following prior studies (Silver et al., 2016; 2017b; Ruoss et al., 2024). For future world representation, we consider a variety of alternative variants, such as purely future actions (denoted as s-aa), action-states (denoted as s-asa), and action-state-values (denoted as s-avsav, etc. We analyze the performance of different future paradigms in Section §4.3. The s-asa approach is ultimately chosen as our modeling paradigm considering the effectiveness and simplicity. The policy distribution at state $s_i$ considering the future is given by:

$$p_{\boldsymbol{\theta}}(a_i, s_{i+1}, a_{i+1}, \ldots, s_{i+h-1}, a_{i+h-1}|s_i), \tag{2}$$

where $h > 1$ is the future horizon.

## 3.2 TRAINING

In order to train a policy that models Eq.(2), we consider a supervised training approach leveraging Stockfish (Romstad et al., 2008). We utilize Stockfish 16, currently the world's strongest search-based engine, as an oracle to label board states extracted from randomly selected games on lichess.org. We approximate the optimal policy $\pi^*$ with $\pi^{SF}$ and obtain each action by taking $a_j^{SF} = \arg\max_{a_j} Q^{SF}(s_j, a_j)$. For a given world horizon $h$, we construct a dataset $\mathcal{D} = \{(s_i, (a_i^{SF}, s_{i+1}, a_{i+1}^{SF}, \ldots, s_{i+h-1}, a_{i+h-1}^{SF}))\}$, where the oracle future path means playing some move that has the maximum evaluation for the best opponent's reply for both players.

An intuitive way to use $\mathcal{D}$ is to train a network to directly predict the entire concatenated next action and future sequence $a_i^{SF} \parallel z_i^{SF}(z_i^{SF} := s_{i+1} \parallel a_{i+1}^{SF} \parallel \cdots \parallel s_{i+h-1} \parallel a_{i+h-1}^{SF})$. Nonetheless, we observe that this approach not only fails to predict the future but also impedes the learning of the next action $a_i^{SF}$ (see Section §4.3). Therefore, we resort to diffusion modeling (Sohl-Dickstein et al., 2015) as a powerful sequence modeling approach with strong expressive capabilities. The bidirectional multi-layer self-attention and iterative denoising mechanism are expected to enhance the prediction of the next action by considering future information. Specifically, we consider discrete diffusion modeling and streamline $L_{\text{vb}}$ in Eq.(1) into a weighted cross-entropy loss motivated by Austin et al. (2021); Zheng et al. (2023); Shi et al. (2024); Sahoo et al. (2024). The KL term $D_{\text{KL}}[q(\boldsymbol{x}_{t-1}|\boldsymbol{x}_t, \boldsymbol{x}_0)||p_{\boldsymbol{\theta}}(\boldsymbol{x}_{t-1}|\boldsymbol{x}_t)$ for each individual random variable is simplified as $-\lambda_t 1_{\boldsymbol{x}_t \neq \boldsymbol{x}_0} \boldsymbol{x}_0^\top \log f(\boldsymbol{x}_t; \boldsymbol{\theta})$, where and $L_{\text{vb}}$ becomes:

$$L_{\text{vb}} = -\mathbb{E}_{q(\boldsymbol{x}_0)} \sum_{t=1}^{T} \lambda_t \mathbb{E}_{q(\boldsymbol{x}_t|\boldsymbol{x}_0)} 1_{\boldsymbol{x}_t \neq \boldsymbol{x}_0} \boldsymbol{x}_0^\top \log f(\boldsymbol{x}_t; \boldsymbol{\theta}), \tag{3}$$

where $\lambda_t = \frac{\alpha_{t-1} - \alpha_t}{1 - \alpha_t} \in (0, 1]$ is a time-dependent reweighting term that assigns lower weight for noisier $\boldsymbol{x}_t$, and $\alpha_t \in [0, 1]$ belongs to a predefined noise scheduler that controls the level of noise in $\boldsymbol{x}_t$ at timestep $t$. We explore multiple variants of $\lambda_t$ in Section §4.3. To enable conditional training with a given state, we freeze the state tokens and perform denoising on the next action $a_i^{SF}$ and all futures tokens $z_i^{SF}$. We employ Monte Carlo sampling with regard to $\boldsymbol{x}_0$, $\boldsymbol{x}_t$ and $t$ when optimizing $L_{\text{vb}}$. We provide detailed derivations in Appendix B.2. We elaborate the training procedure in Algorithm 1.

## 3.3 INFERENCE

During inference, taking $\arg\max_{a_i} p_{\boldsymbol{\theta}}(a_i|s_i)$ as in one-step policy in DIFFUSEARCH requires marginalizing over all future with horizon $h$, i.e., $p_{\boldsymbol{\theta}}(a_i|s_i) = \sum_{z_i} p_{\boldsymbol{\theta}}(a_i, z_i|s_i)$, which is intractable due to the exponential-growing search space when $h$ goes larger, e.g., the game tree contains $b^h$ nodes and the branching factor $b$ is around 31 on average in chess (Barnes, 2019). One simplified approach to comparing actions is to measure the best future if one action is taken, which can be reflected by the joint probability $p_{\boldsymbol{\theta}}(a_i, z_i|s_i)$. Therefore, we resort to $\arg\max_{a_i, z_i} p_{\boldsymbol{\theta}}(a_i, z_i|s_i)$, which does not involve marginalization and can be achieved by sampling from the trained model. During diffusion sampling, we adopt an easy-first decoding strategy (Savinov et al., 2021; Chang et al., 2022), which achieves better performance compared to the random decoding approach employed by Austin et al. (2021). Specifically, at diffusion timestep $t$, the tokens within the least $100 * \frac{t-1}{T}\%$ predictive log-likelihood are selected to be reset to the noise state. To change search depth, we mainly train separate models on $\mathcal{D}$ with different $h$, and study a single model on $\mathcal{D}$ with mixed $h$ in Appendix C.1. We elaborate the inference algorithm in Algorithm 2.

## 4 EXPERIMENTS

## 4.1 SETUP

**Baselines** We compare our model with three Transformer models proposed in Ruoss et al. (2024): State-action model (S-A) which learns to predict next move via behavioral cloning; State-value model (S-V) which predicts next move via comparing the value of next states; and Action-value model (SA-V) which predicts next move via comparing the value of each legal actions at the current state. We also integrate the trained S-A and S-V models into MCTS following AlphaZero (Silver et al., 2017a).

**Algorithm 1** DIFFUSEARCH Training

**Input:** dataset $\mathcal{D} = \{(s, (a, z))\}$, neural network $f(\cdot; \boldsymbol{\theta})$, timesteps $T$.
**Output:** model parameters $\boldsymbol{\theta}$.
Denote state length $l = |s|$;
**repeat**
 Draw $(s, (a, z)) \sim \mathcal{D}$ and obtain $\mathbf{x}_{0,1:N} = s \,||\, a \,||\, z$ ($||$: concat);
 Draw $t \in \text{Uniform}(\{1, \ldots, T\})$;
 Draw $\mathbf{x}_{t,n} \sim q(\mathbf{x}_{t,n}|\mathbf{x}_{0,n})$ for $n \in \{l+1, \ldots, N\}$;
 $L(\boldsymbol{\theta}) = -\lambda_t \sum_{n=l+1}^{N} \mathbf{1}_{\mathbf{x}_{t,n} \neq \mathbf{x}_{0,n}} \mathbf{x}_{0,n}^{\top} \log f(\mathbf{x}_{t,n}; \boldsymbol{\theta})$;
 Minimize $L(\boldsymbol{\theta})$ with respect to $\boldsymbol{\theta}$;
**until** converged

**Algorithm 2** DIFFUSEARCH Inference

**Input:** board state $s$, trained network $f(\cdot; \boldsymbol{\theta})$, timesteps $T$.
**Output:** next action $a$.
Denote state length $l = |s|$;
Initialize $\mathbf{x}_{T,1:l} = s$ and $\mathbf{x}_{T,l+1:N} \sim q_{\text{noise}}$;
**for** $t = T, \ldots, 1$ **do**
 **for** $n = l+1, \ldots, N$ **do**
  Draw $\widetilde{\mathbf{x}}_{0,n} \sim \text{Cat}(f(\mathbf{x}_{t,n}; \boldsymbol{\theta}))$;
  Draw $\mathbf{x}_{t-1,n} \sim q(\mathbf{x}_{t-1,n} \mid \mathbf{x}_{t,n}, \widetilde{\mathbf{x}}_{0,n})$;
 **end for**
**end for**
**Return** $a = \mathbf{x}_{0,l+1}$.

**Data** We construct a dataset for supervised training by downloading games from lichess recorded in February 2023. When analyzing the scaling behavior, we use up to 100k games, while reverting to the default 10k games for other experiments due to resource constraints. We show the data statistics in Table 1. Following Ruoss et al. (2024), we convert the centipawns returned by Stockfish to the win percentage and then discretize it into 128 bins to represent

Table 1: Data statistics.

| Stage | Records | Games |
|---|---|---|
| Train SA-V (100k) | 193,189,573 | 100,000 |
| Train SA-V (10k) | 17,448,268 | 10,000 |
| Train others (100k) | 6,564,661 | 100,000 |
| Train others (10k) | 659,576 | 10,000 |
| Action Test | 62,561 | 1,000 |
| Puzzle Test | 36,816 | 10,000 |

value in S-V and SA-V. We encode the state as a fixed-length FEN string with 77 characters by padding with '.' if needed. Actions are stored in UCI notation with 1968 possible moves in total. We provide example training data for each paradigm in Appendix C.3.

**Implementation Details** For all the neural models in this paper, we use the same decoder-only GPT-2 transformer architecture (Vaswani et al., 2017; Radford et al., 2019) for a rigorous comparison. For DIFFUSEARCH, we convert casual attention into full attention without introducing additional learned parameters. We train all baseline models until convergence and set a maximum of 200 epochs for diffusion models due to their slow convergence. We use the Adam optimizer (Kingma & Ba, 2015), a learning rate of 3e-4, and a batch size of 1024 for all models. By default, we set the horizon $h$ to be 4, the number of network layers to be 8 (with a total parameter size of 7M), the diffusion timesteps to be 20, and an absorbing noise type. By default, 100 simulations are utilized in MCTS-enhanced policy, and its impact is analyzed in Figure 3. We adjust $c_{\text{puct}}$ and $\tau$, constants determining the level of exploration in MCTS, on a held-out set and set them to $c_{\text{puct}} = 0.1$ and $\tau = 1$ for its superior performance. All experiments are done on 8 NVIDIA V100 32G GPUs.

**Evaluation Metrics** We mainly consider three metrics to evaluate the policies following (Ruoss et al., 2024): 1) **Action Accuracy**: the percentage of the test set in which the model selects the same action as the ground truth; 2) **Puzzle Accuracy**: the percentage of puzzles where the policy's action sequence exactly matches the known solution action sequence and we use 10k puzzles with difficulty rated by Elo from 399 to 2867 provided by (Ruoss et al., 2024); 3) **Tournament Elo**: the Elo ratings calculated using BayesElo (Coulom, 2008) in an internal tournament involving all policies, where each pair of policies played 400 games, resulting in a total of 6000 games.

## 4.2 MAIN RESULTS

We report the prediction and playing strength comparison for our model against baselines in Table 2. Additionally, we report the performance of Stockfish 16 with a time limit of 0.05s per legal move, which stands as the oracle used to generate our dataset. We find DIFFUSEARCH significantly outperforms the S-A model by 653 Elo and 19% action accuracy, indicating the effectiveness of DIFFUSEARCH in improving next action prediction through future prediction. Remarkably, despite utilizing 20 times fewer data records than the SA-V model, our model demonstrates superior performance with approximately 10% higher action accuracy. Our model demonstrates superior performance over the MCTS-based agent by achieving a higher Elo difference of 542 and an increased action accuracy of 14%. This highlights the effectiveness of DIFFUSEARCH in modeling multi-step

Table 2: Prediction and playing strength comparison for our model against baselines and the oracle Stockfish 16. The `S-V` and `SA-V` models can be seen as depth-one search. The best results are bold.

| Agent | Search | Tournament Elo | Accuracy | |
|---|---|---|---|---|
| | | | **Puzzles** | **Actions** |
| Stockfish 16 (0.05s) [oracle] | ✓ | 2689 | 99.10 | 100.00 |
| *10k training games* | | | | |
| Transformer (S-A) | | 1075 | 3.95 | 22.10 |
| Transformer (S-V) | ✓ | 1028 | 12.20 | 21.45 |
| Transformer (SA-V) | ✓ | 1294 | 12.74 | 31.50 |
| Transformer (100 MCTS simulations) | ✓ | 1186 | 6.85 | 27.34 |
| DIFFUSEARCH (Ours) | | **1728** | **39.49** | **41.31** |
| *100k training games* | | | | |
| Transformer (S-A) | | 1467 | 20.83 | 36.58 |
| Transformer (S-V) | ✓ | 1078 | 17.42 | 28.89 |
| Transformer (SA-V) | ✓ | 1521 | 24.25 | 39.76 |
| Transformer (100 MCTS simulations) | ✓ | 1469 | 20.71 | 38.05 |
| DIFFUSEARCH (Ours) | | **1995** | **58.46** | **48.66** |

simulations when compared with the step-by-step MCTS-enhanced policy, which relies on a robust value model and necessitates a careful balance between the policy and value models.

## 4.3 ABLATIONS

**Future paradigm matters** We compare baselines and different future paradigms during the training of DIFFUSEARCH with horizon $h = 4$ in Table 3. For each future paradigm, we compare training with autoregressive Transformer and DIFFUSEARCH. We find that directly performing future action prediction (`S-AA`) (Chi et al., 2023) with DIFFUSEARCH hurts performance compared to `S-A` due to the difficulty of future move prediction in chess. However, after we integrate future states, we observe significant performance improvements when comparing `S-AA` (15.07) to `S-ASA` (41.31), and also when comparing `S-AVAV` (17.63) to `S-AVSAV` (40.69). No further

Table 3: Action accuracy comparison of baselines and different future paradigms.

| Paradigms | Transformer | DIFFUSEARCH |
|---|---|---|
| S-A | 22.10 | - |
| S-V | 21.45 | - |
| SA-V | 31.50 | - |
| S-AA | 26.62 | 15.07 |
| S-ASA | 27.39 | 41.31 |
| S-ASS | 24.93 | 41.19 |
| S-AVAV | 25.92 | 17.63 |
| S-AVSAV | 25.59 | 40.69 |

improvement is observed when integrating the values in DIFFUSEARCH, which may be attributed to training on the optimal future trajectory rather than all possible trajectories. For training using autoregressive Transformer, we observe that the overall performance hovers around 26%. This performance level is superior to that achieved by `S-A` (22.1%), attributed to the utilization of more (S, A) pairs (e.g., each `S-ASA` record containing $h$ (S, A) pairs). However, the performance falls short when compared to DIFFUSEARCH, which underscores the importance of modeling bidirectional context to leverage future information for subsequent action prediction.

**Ensuring the validity of future world dynamics is crucial** After we discuss the future paradigm, we now investigate the effect of future quality on performance, as shown in Table 4. For better illustration, denote a sequence of future horizon 2 as $[s_1 = f(s_0, a_0), a_1 = g(s_1), s_2 = f(s_1, a_1), a_2 = g(s_2)]$, where $f$ is a world dynamic function and $g$ is a policy function. $s_0$ is the current state and $a_0$ is the move suggested by Stockfish. We first utilize random state-action sequences for future steps, where both actions and states were randomly selected (i.e., random world $f$ and random policy $g$). This methodology did not yield performance enhancements. Subsequently, we explore selecting random actions and incorporating the resulting state

Table 4: Future world quality in supervising the model for the S-ASA paradigm.

| Future Quality | Acc. |
|---|---|
| Without future | 22.10 |
| + Random world+policy | 22.69 |
| + Random policy | 39.47 |
| + Stockfish policy | 41.31 |

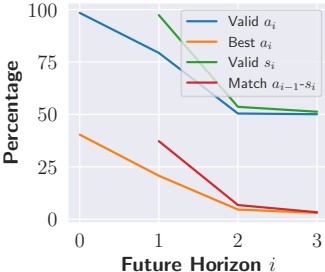 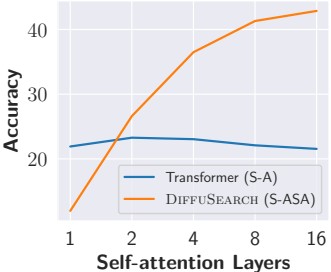 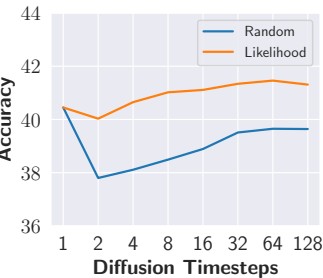

Figure 2: **(Left)** Prediction quality analysis for DIFFUSEARCH at different future steps. **(Middle)** Action accuracy when scaling self-attention layers. **(Right)** Action accuracy when increasing diffusion timesteps.

from executing those actions (i.e., random policy $g$ but an oracle world $f$), which notably outperforms the initial strategy. This underscores the significance of aligning states with corresponding actions, mirroring the dynamics of the world. Finally, we investigate incorporating high-quality future actions suggested by Stockfish (i.e., Stockfish $g$ and oracle world $f$) and observe additional performance improvements compared to the random action selection approach.

**Proper discrete diffusion modeling helps** Given the dataset $\mathcal{D}$ annotated with future states and actions, we investigate alternative ways to train the model, as presented in Table 5. We first observe it is hard to teach the model to directly output the entire future sequence, leading to lower performance compared to auto-regressive training. Secondly, we employ continuous Gaussian diffusion VDM (Kingma et al., 2021) and observe its superior performance compared to the Direct and auto-regressive methods, but inferior compared to discrete approaches. The absorbing diffusion with reciprocal $\lambda_t = 1/t$ obtained by setting $\alpha_t = 1 - \frac{t}{T}$ in Eq.(3) is a simplified expression from D3PM (Austin et al., 2021), which we find significantly outperforms continuous diffusion. Fi-

Table 5: Comparison of training methods. Direct: train the model to predict the entire future sequence at once.

| Method | Acc. |
|---|---|
| Direct | 20.61 |
| Auto-regressive | 27.39 |
| Gaussian | 31.91 |
| Absorbing, $\lambda_t = 1$ | 39.66 |
| Absorbing, $\lambda_t = 1/t$ | 39.07 |
| Absorbing, $\lambda_t = 1 - \frac{t-1}{T}$ | 41.31 |
| Multinomial, $\lambda_t = 1 - \frac{t-1}{T}$ | 40.08 |

nally, we discover a linear $\lambda_t$ (Bond-Taylor et al., 2022; Zheng et al., 2023) further exceeds the constant and reciprocal ones, as well as the multinomial counterpart.

## 4.4 ANALYSIS

**Does DIFFUSEARCH predict accurate future information?** We analyze the percentage of valid actions and the optimal action recommended by Stockfish for each predicted action. The best $a_0$ metric is exactly the action accuracy by definition.. Additionally, we assess whether each predicted state is a valid representation and if $s_i$ corresponds to the resulting state when action $a_{i-1}$ is taken at $s_{i-1}$. The initial state $s_0$ provided as input is excluded, and the results are presented in the left figure of Figure 2. We observe that the first action, denoted as $a_0$, are almost 100% valid. As we progress through future steps, both the valid rate and the optimal rate decline. However, even at $i = 3$, where the valid rate stands at 50%, it surpasses the random move baseline of approximately 1.6% (calculated as the average number of legal actions per move, 31, divided by the total number of moves, 1968). This indicates that the model retains a certain level of predictive capability for future moves, albeit with reduced performance. A similar pattern appears in the evaluation of states, where the accuracy is perfect for the first predicted state $s_1$ but diminishes in subsequent predictions. In Appendix Table 8, we demonstrate that further increasing the training data enhances the effectiveness of the world model within DIFFUSEARCH, achieving over 90% accuracy in predicting valid and matched future states corresponding to the preceding action.

**How does DIFFUSEARCH leverage future information?** We attributes the future-aware ability of DIFFUSEARCH mainly to self-attention and iterative decoding process, as shown in the middle and right figures of Figure 2, respectively. When employing a single self-attention layer, our model

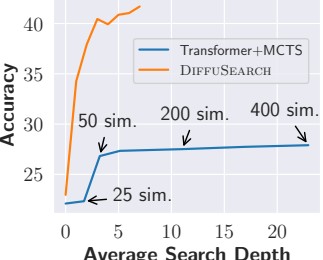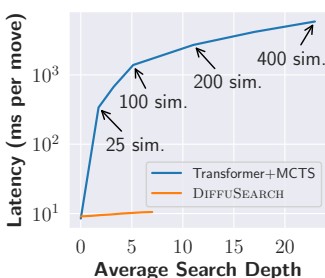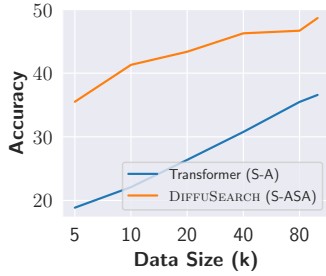

Figure 3: **(Left)** Action accuracy when increasing average search depth in MCTS through more simulations and DIFFUSEARCH through context length extension. **(Middle)** Latency measured by ms per second when increasing search depth. **(Right)** Action accuracy when scaling data size.

exhibits inferior performance compared to the `S-A` model, yet surpasses it with two layers. Moreover, its performance steadily enhances as we augment the number of layers. This suggests that with additional layers, there is more chance for the subsequent actions and future to interact reciprocally, akin to the enhancement in the action prediction with increased MCTS simulations. We do not observe a similar upward trend in the performance of `S-A` model when increasing attention layers as in (Ruoss et al., 2024), possibly indicating that the available data (10k) does not necessitate the integration of more layers. In the right figure of Figure 2, it is evident that employing an appropriate decoding strategy (such as likelihood-based) further enhances next-action prediction as the number of iterations grows. However, the overall improvement is relatively modest compared to increasing the attention layers.

**Explicit search vs. Implicit search**   Based on our previous analysis, we can consider DIFFUSE-ARCH as performing implicit search through the inner self-attention layers and the multi-step diffusion process. Now, we aim to evaluate the efficiency and effectiveness of this implicit search in comparison to explicit search using MCTS when conducting deeper searches. In DIFFUSEARCH, deeper search is realized by increasing the context length (80 tokens per search depth), whereas in MCTS, it is achieved through running more simulations. In the left figure of Figure 3, it is evident that DIFFUSEARCH exhibits significant enhancement when increasing search depth, while MCTS becomes stagnant after 50 simulations at a search depth of around 4. This could be attributed to the accumulated errors caused by the value network due to a limited horizon. In the middle figure of Figure 3, we measure the latency per move for Transformer with MCTS and DIFFUSEARCH on a single V100 GPU with batch size 1. The performance of Transformer combined with MCTS is notably affected by the necessity of invoking the value network for every simulation. In contrast, DIFFUSEARCH experiences only a slight rise in latency as it requires just one call for greater depth.

**Scaling**   In Figure 2, the effectiveness of model scaling in DIFFUSEARCH has been observed. Here we explore the impact of increasing the dataset size on the performance. Specifically, we conduct experiments training the DIFFUSEARCH `S-ASA` model with a horizon of 4 and the Transformer `S-A` using game sizes ranging from 5k to 100k, as shown in the right figure of Figure 3. Both the Transformer and DIFFUSEARCH models exhibit a log-2 scaling behavior, showing that doubling the training data results in a linear increase in accuracy. Scaling also enhances future prediction significantly, leading to a more valid and accurate representation of future actions and states, as well as a near-perfect level of capturing the state-action transition dynamics, as detailed in Appendix C.2.

**Case study**   We sample several challenging puzzles from Lichess (with Elo ratings above 1800) to compare the predictions of DIFFUSEARCH and Transformer (`S-A`). Two instances are shown in Figure 4, with additional cases provided in Appendix C.4. DIFFUSEARCH demonstrates superior foresight, accurately predicting critical exchanges and piece sacrifices that lead to long-term strategic advantages. In the left puzzle, DIFFUSEARCH strategically sacrifices the rook to set up a long-term checkmate situation against the opponent. This maneuver compels the opponent to defend and creates an opportunity to capture the queen, facilitating valuable piece exchanges. The `S-A` model, unfortunately, makes a critical error by focusing on achieving direct checkmate without considering the possibility of the opponent's queen launching a counterattack. Similarly, in the right puzzle, DIFFUSEARCH anticipates an exchange sacrifice, correctly valuing the long-term positional benefits

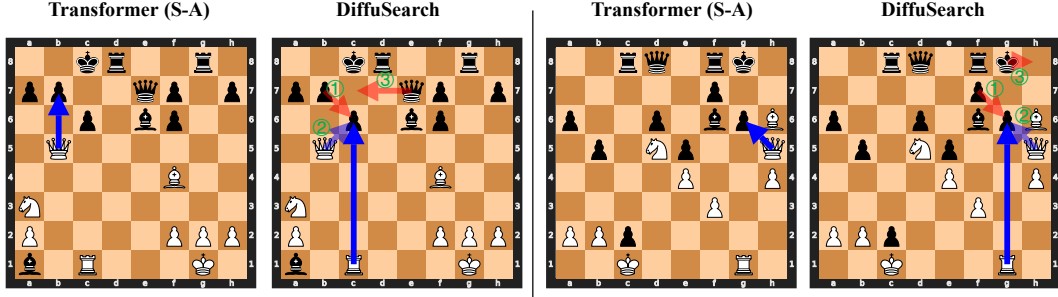

Figure 4: Two examples of Transformer (`S-A`) and DIFFUSEARCH solving challenging puzzles. The predicted next move is in blue for both policies. The predicted future actions from DIFFUSEARCH are in light blue and red representing the two players, respectively, along with the numerical counters 1, 2, and 3 indicating future steps.

of opening lines by sacrificing the rook for its queen. Conversely, the `S-A` model misjudges the value of this exchange, leading to suboptimal moves. These findings highlight the effectiveness of DIFFUSEARCH in long-term planning without relying on explicit search.

## 5 RELATED WORK

### 5.1 NEURAL NETWORKS FOR CHESS

The development of chess AI has undergone a significant transformation, shifting from the explicit design of search strategies and heuristics to the more data-driven and learning-based approaches. The early research, exemplified by Turing's investigations (Burt, 1955) and NeuroChess (Thrun, 1994), heavily depended on handcrafted search algorithms and heuristics, eventually leading to the development of powerful search engines like Deep Blue (Campbell et al., 2002) and Stockfish (Romstad et al., 2008). However, the emergence of neural network-based approaches, typically AlphaZero (Silver et al., 2017a), marked a paradigm shift, where deep reinforcement learning equipped with Monte Carlo Tree Search (MCTS) enabled the system to learn its own heuristics, i.e., the policy and value networks, without the need for manual design (Klein, 2022; McGrath et al., 2021). The rise of large language models (LLMs) has also inspired innovations in chess AI, such as the evaluation (Toshniwal et al., 2022; Carlini, 2023) and interpretation (Li et al., 2023a; Karvonen, 2024) of LLMs' ability to play chess, the integration of chess-related text data into training (Feng et al., 2024), and the exploring of searchless models by scaling the policy networks (Ruoss et al., 2024). Despite this, lookahead search methods like beam search (Feng et al., 2024) and even depth-one search with the value network (Ruoss et al., 2024) remain superior to the policy models as action predictors, which is the same as in the AlphaZero era (Silver et al., 2017b; Team., 2018). This underscores the continued significance of lookahead information for move prediction in chess. In contrast to prior research, we explore directly teaching the policy model to look ahead, thereby eliminating the requirement of handcrafted search algorithms or separate value networks.

### 5.2 DIFFUSION MODELS

Diffusion models (Sohl-Dickstein et al., 2015; Ho et al., 2020; Austin et al., 2021), a powerful class of generative models, have been applied to various fields such as image generation (Dhariwal & Nichol, 2021; Rombach et al., 2022; Croitoru et al., 2023, *inter alia*), text generation (Li et al., 2022; Gong et al., 2022; Zheng et al., 2023; Lou et al., 2023; Li et al., 2023b, *inter alia*) and reinforcement learning (Janner et al., 2022; Ajay et al., 2022; Chi et al., 2023; Zhu et al., 2023, *inter alia*). Theoretically, diffusion models perform a multi-step denoising process to progressively convert a random noise into a data sample, and the denoising procedure can be seen as parameterizing the gradients of the data distribution (Song & Ermon, 2019), connecting them to score matching (Hyvärinen & Dayan, 2005) and energy-based models (LeCun et al., 2006). Particularly, diffusion models have demonstrated their effectiveness in tasks that require global control and future planning, such as math reasoning (Ye et al., 2024b; Gong et al., 2024), paragraph generation (Zhang et al., 2023b),

trajectory planning (Janner et al., 2022) and robot manipulation (Chi et al., 2023). The superior planning ability of diffusion compared to autoregression has been studied in Ye et al. (2024a). Compared to Diffusion Policy (Chi et al., 2023), DIFFUSEARCH internalizes a world model inside the policy, which we find is crucial in Section §4.3. Furthermore, we focus on exploring diffusion models for implicit search as an alternative to the one-step policy with explicit search to deal with complex tasks that require search.

### 5.3 WORLD MODELS

The primary goal of a world model is to capture the underlying dynamics of the environment and predict the future outcome of certain actions in the context of model-based reinforcement learning (MBRL) (Wang et al., 2019; Moerland et al., 2023). The learned world model can be used for policy optimization of a RL agent (Sutton, 1991; Feinberg et al., 2018; Hafner et al., 2020a) and allow the agent to explicitly reason about the future consequences of its actions (Hafner et al., 2019; Schrittwieser et al., 2020; Ye et al., 2021). Most of the conventional world models (Hafner et al., 2020a;b; 2023) rely on single-step prediction, which suffer from compounding errors (Asadi et al., 2019; Xiao et al., 2019; Lambert et al., 2022). Recently, there has been growing interest in building multi-step world models utilizing diffusion models (Zhang et al., 2023a; Rigter et al., 2023; Jackson et al., 2024; Ding et al., 2024), which, however, separate the world model and policy. Similar to ours, Diffuser (Janner et al., 2022) and Decision Diffuser (DD; Ajay et al. 2022) also unify the world model with the policy. However, the modeling details, training paradigm, and action prediction differ. Specifically, both of them employ continuous diffusion while we use discrete diffusion. In addition, Diffuser trains an unconditioned model and requires a guidance function to obtain desired actions, while we model the best action and future trajectory condition on a given state. DD models state-only future trajectories and predicts the action through an inverse dynamics model while we model both future states and actions. Finally, the comparison of diffusion world model and explicit search has not been rigorously explored in domains that require precise and sophisticated lookahead such as chess, to the best of our knowledge.

## 6 CONCLUSION AND DISCUSSION

In this study, we present evidence showcasing the potential transition from employing explicit search on a one-step policy to implicit search within a future-aware policy on the classic board game Chess. The proposed model, DIFFUSEARCH, demonstrates not only superior performance compared to the searchless policy but also the policy empowered by explicit search. We provide extensive experiments to demonstrate and analyze DIFFUSEARCH. More broadly, the ideas and techniques discussed in this controlled task may eventually be valuable in natural language settings to improve the current next-token prediction LLMs as well.

We now discuss some limitations and workarounds in our study. Firstly, one usage of explicit search such as MCTS is to enhance policy performance through self-play training, such that is able to achieve amazing performance without any human supervision (Silver et al., 2017b). However, our model currently relies on an oracle (Stockfish) to provide future supervision. The integration of DIFFUSEARCH with self-play is an interesting direction to explore. Secondly, our model achieves a deeper search by increasing the context length, with the current training limited to a depth of 7, corresponding to a context length of 648. For scenarios requiring more tokens to represent a state or deeper searches, integrating techniques for long-context models may be useful for efficient training or inference (Dao et al., 2022; Gu & Dao, 2023; Xiong et al., 2024; An et al., 2024). Finally, our model's performance is currently constrained by the relatively small training dataset of up to 100k games due to resource restrictions, considerably less than the 10 million games used in the study by Ruoss et al. (2024). Continuing to scale the model and data remains a valuable direction.

### ACKNOWLEDGMENTS

This research was supported in part by the joint research scheme of the National Natural Science Foundation of China (NSFC) and the Research Grants Council (RGC) under grant number N_HKU714/21.

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

## A    DETAILS ABOUT MCTS-ENHANCED POLICY

This baseline is fully aligned with the approach used in AlphaZero (Silver et al., 2017a). The one-step policy directly predicts the next action, while the MCTS-enhanced Policy constructs a search tree that simulates the future to enhance the evaluation of potential next actions. Each node $s$ in the search tree contains edges $(s, a)$ for all legal actions $a \in \mathcal{A}(s)$. Each edge stores a set of statistics,

$$\{N(s, a), W(s, a), Q(s, a), P(s, a)\}, \tag{4}$$

where $N(s, a)$ is the visit count, $W(s, a)$ is the total action-value, $Q(s, a)$ is the mean action-value, and $P(s, a)$ is the prior probability of selecting that edge. The algorithm proceeds by iterating over the former three phases below and then selects a move to play:

**Selection.**    The algorithm begins at the root node and traverses the tree, selecting child nodes based on strategies to maximize the exploration of promising paths. Specifically, at each intermediate node, an action is selected according to the statistics in the search tree, $a_t = \underset{a}{\arg\max} \left( Q(s_t, a) + U(s_t, a) \right)$, using a variant of the PUCT algorithm,

$$U(s, a) = c_{\text{puct}} P(s, a) \frac{\sqrt{\sum_b N(s, b)}}{1 + N(s, a)}, \tag{5}$$

where $c_{\text{puct}}$ is a constant determining the level of exploration; this search control strategy initially prefers actions with high prior probability and low visit count, but asymptotically prefers actions with high action-value.

**Expansion and evaluation.**    Upon reaching a leaf node, if it does not represent a terminal state (i.e., the end of the game), one or more new child nodes are expanded and evaluated by the policy and value model. The leaf node $s_L$ is added to a queue for neural network evaluation, $v = v_\theta(s_L)$ and $p = p_\theta(s_L)$. The leaf node is expanded and each edge $(s_L, a)$ is initialized to $\{N(s_L, a) = 0, W(s_L, a) = 0, Q(s_L, a) = 0, P(s_L, a) = p_a\}$; the value $v$ is then backed up.

**Backup.**    The edge statistics are updated in a backward pass through each step $t \le L$. The visit counts are incremented, $N(s_t, a_t) = N(s_t, a_t) + 1$, and the action-value is updated to the mean value, $W(s_t, a_t) = W(s_t, a_t) + v, Q(s_t, a_t) = \frac{W(s_t, a_t)}{N(s_t, a_t)}$.

**Play.**    After iteratively cycling through the above phases, a move is selected to play in the root position $s_0$ at the end of the search based on the statistical information, e.g., proportional to its exponentiated visit count, $\pi(a|s_0) = N(s_0, a)^{1/\tau} / \sum_b N(s_0, b)^{1/\tau}$, where $\tau$ is a temperature parameter that controls the level of exploration. The search tree is reused at subsequent time-steps: the child node corresponding to the played action becomes the new root node; the subtree below this child is retained along with all its statistics, while the remainder of the tree is discarded.

## B    DERIVATIONS

### B.1    DISCRETE DIFFUSION

In this section, we provide a detailed derivation of the representation for distributions used in the objective Eq.(1), which we bring here for a better illustration:

$$L_{\text{vb}} = \mathbb{E}_{q(\boldsymbol{x}_0)} \Big[ \underbrace{D_{\text{KL}}[q(\boldsymbol{x}_T|\boldsymbol{x}_0)||p(\boldsymbol{x}_T)]}_{L_T} + \sum_{t=2}^{T} \underbrace{\mathbb{E}_{q(\boldsymbol{x}_t|\boldsymbol{x}_0)} \big[ D_{\text{KL}}[q(\boldsymbol{x}_{t-1}|\boldsymbol{x}_t, \boldsymbol{x}_0)||p_{\boldsymbol{\theta}}(\boldsymbol{x}_{t-1}|\boldsymbol{x}_t)] \big]}_{L_{t-1}} \\ \underbrace{-\mathbb{E}_{q(\boldsymbol{x}_1|\boldsymbol{x}_0)}[\log p_{\boldsymbol{\theta}}(\boldsymbol{x}_0|\boldsymbol{x}_1)]}_{L_0} \Big].$$

where $L_T$ is a constant when a fixed prior $p(\boldsymbol{x}_T)$ is employed. In discrete diffusion, both the forward and backward distribution are defined as categorical distribution, e.g., $q(\boldsymbol{x}_t|\boldsymbol{x}_{t-1}) = \text{Cat}(\boldsymbol{x}_t; \boldsymbol{p} = \boldsymbol{Q}_t^\top \boldsymbol{x}_{t-1})$ and $p_{\boldsymbol{\theta}}(\boldsymbol{x}_{t-1}|\boldsymbol{x}_t) = q(\boldsymbol{x}_{t-1}|\boldsymbol{x}_t, f(\boldsymbol{x}_t; \boldsymbol{\theta}))$ (Hoogeboom et al., 2021), where $\boldsymbol{Q}_t$ is a pre-defined $K \times K$ transition matrix and $K$ is the size of categories.

**The posterior $q(\boldsymbol{x}_{t-1}|\boldsymbol{x}_t, \boldsymbol{x}_0)$** Starting from $\boldsymbol{x}_0$, we obtain the following $t$-step marginal and posterior at time $t-1$:

$$q(\boldsymbol{x}_t|\boldsymbol{x}_0) = \text{Cat}\left(\boldsymbol{x}_t; \boldsymbol{p} = \overline{\boldsymbol{Q}}_t^\top \boldsymbol{x}_0\right), \quad \text{with} \quad \overline{\boldsymbol{Q}}_t = \boldsymbol{Q}_1 \boldsymbol{Q}_2 \ldots \boldsymbol{Q}_t$$

$$q(\boldsymbol{x}_{t-1}|\boldsymbol{x}_t, \boldsymbol{x}_0) = \frac{q(\boldsymbol{x}_t|\boldsymbol{x}_{t-1}, \boldsymbol{x}_0)q(\boldsymbol{x}_{t-1}|\boldsymbol{x}_0)}{q(\boldsymbol{x}_t|\boldsymbol{x}_0)} = \text{Cat}\left(\boldsymbol{x}_{t-1}; \boldsymbol{p} = \frac{\boldsymbol{Q}_t \boldsymbol{x}_t \odot \overline{\boldsymbol{Q}}_{t-1}^\top \boldsymbol{x}_0}{\boldsymbol{x}_t^\top \overline{\boldsymbol{Q}}_t^\top \boldsymbol{x}_0}\right), \quad (6)$$

where $q(\boldsymbol{x}_t|\boldsymbol{x}_{t-1}, \boldsymbol{x}_0) = q(\boldsymbol{x}_t|\boldsymbol{x}_{t-1})$ due to the Markov property of the forward process. The KL divergence between $q$ and $p_{\boldsymbol{\theta}}$ can be computed by simply summing over all possible values of each random variable. The cumulative products $\overline{\boldsymbol{Q}}_t$, which can be computed in closed form or precomputed for all $t$ depending on the choice $\boldsymbol{Q}_t$, may be prohibitive for large $T$ and number of categories. Therefore, two commonly used forms of $\boldsymbol{Q}$ are introduced by Hoogeboom et al. (2021) and Austin et al. (2021), which ensures $\overline{\boldsymbol{Q}}_t$ can still be computed efficiently, allowing the framework to scale to a larger number of categories.

**Multinominal diffusion** The transition matrix initially proposed for the binary scenario by Sohl-Dickstein et al. (2015) and later expanded to categorical by Hoogeboom et al. (2021) can be represented as a $K \times K$ matrix:

$$[\boldsymbol{Q}_t]_{ij} = \begin{cases} 1 - \frac{K-1}{K}\beta_t & \text{if} \quad i = j \\ \frac{1}{K}\beta_t & \text{if} \quad i \neq j \end{cases}.$$

This transition matrix can also be written as $(1 - \beta_t)I + \beta_t \mathbf{1}\mathbf{1}^\top/K$, where $\mathbf{1}$ is a column vector of all ones. The transition matrices $\overline{\boldsymbol{Q}}$ can be computed in closed form. Denote the vector represents the uniform noise distribution as $q_{\text{noise}} = \mathbf{1}/K$. In each step, we transition to another token with probability $\beta_t$ and stay the same with probability $1 - \beta_t$. After $t$ steps, the only operative quantity is the probability of not yet having transitioned to another token, given by $\alpha_t = \prod_{i=0}^t (1 - \beta_i)$. Therefore, we derive:

$$\overline{\boldsymbol{Q}}_t = \alpha_t I + (1 - \alpha_t)\mathbf{1}q_{\text{noise}}^\top, \quad (7)$$

where setting $q_{\text{noise}} = \mathbf{1}/K$ gives the $\overline{\boldsymbol{Q}}_t$ for multinominal diffusion.

**Absorbing diffusion** For diffusion models with an absorbing state $m$, the following matrix is introduced by Austin et al. (2021):

$$[\boldsymbol{Q}_t]_{ij} = \begin{cases} 1 & \text{if} \quad i = j = m \\ 1 - \beta_t & \text{if} \quad i = j \neq m \\ \beta_t & \text{if} \quad j = m, i \neq m \end{cases}.$$

The transition matrix can also be written as $(1 - \beta_t)I + \beta_t \mathbf{1}e_m^\top$, where $e_m$ is a vector with a one on the absorbing state $m$ and zeros elsewhere. Since $m$ is an absorbing state, the corruption process converges not to a uniform distribution but to the point-mass distribution on $m$. For text generation, $m$ is the [MASK] token and this leads to a BERT-like training objective (Devlin et al., 2019), while masks tokens according to some schedule and learns to denoise them iteratively. Similar as in multinomial diffusion, we set $q_{\text{noise}} = e_m$ for absorbing diffusion, where $e_m$ is a one-hot vector on the [MASK] token, and obtain $\overline{\boldsymbol{Q}}_t$ based on Eq.(7).

## B.2 A SIMPLIFIED OBJECTIVE

The categorical distribution parameterized by $\boldsymbol{p}$ for each variable that follows $q(\boldsymbol{x}_{t-1}|\boldsymbol{x}_t, \boldsymbol{x}_0)$ based on Eq.(6) is given as:

$$
\begin{aligned}
\boldsymbol{p} &= \frac{\boldsymbol{Q}_t\boldsymbol{x}_t \odot \overline{\boldsymbol{Q}}_{t-1}^\top \boldsymbol{x}_0}{\boldsymbol{x}_t^\top \overline{\boldsymbol{Q}}_t^\top \boldsymbol{x}_0} \\
&= \frac{[(1-\beta_t)\boldsymbol{x}_t + \beta_t\sigma_{\boldsymbol{x}_t}\mathbf{1}] \odot [\alpha_{t-1}\boldsymbol{x}_0 + (1-\alpha_{t-1})q_{\text{noise}}]}{\alpha_t\boldsymbol{x}_t^\top \boldsymbol{x}_0 + (1-\alpha_t)\boldsymbol{x}_t^\top q_{\text{noise}}} \\
&= \frac{(1-\beta_t)\alpha_{t-1}\boldsymbol{x}_t \odot \boldsymbol{x}_0 + (1-\beta_t)(1-\alpha_{t-1})\boldsymbol{x}_t \odot q_{\text{noise}} + \beta_t\alpha_{t-1}\sigma_{\boldsymbol{x}_t}\mathbf{1} \odot \boldsymbol{x}_0 + \beta_t(1-\alpha_{t-1})\sigma_{\boldsymbol{x}_t}\mathbf{1} \odot q_{\text{noise}}}{\alpha_t\boldsymbol{x}_t^\top \boldsymbol{x}_0 + (1-\alpha_t)\boldsymbol{x}_t^\top q_{\text{noise}}} \\
&= \frac{(1-\beta_t)\alpha_{t-1}\boldsymbol{x}_t \odot \boldsymbol{x}_0 + (1-\beta_t)(1-\alpha_{t-1})\sigma_{\boldsymbol{x}_t}\boldsymbol{x}_t + \beta_t\alpha_{t-1}\sigma_{\boldsymbol{x}_t}\boldsymbol{x}_0 + \beta_t(1-\alpha_{t-1})\sigma_{\boldsymbol{x}_t}q_{\text{noise}}}{\alpha_t\boldsymbol{x}_t^\top \boldsymbol{x}_0 + (1-\alpha_t)\sigma_{\boldsymbol{x}_t}},
\end{aligned}
$$

where $\sigma_{\boldsymbol{x}_t} := q_{\text{noise}}(\boldsymbol{u} = \boldsymbol{x}_t)$ represents the probability of noise drawn from $q_{\text{noise}}$ being equal to $\boldsymbol{x}_t$. Note $\boldsymbol{x}_t \odot \boldsymbol{x}_0 = 0$ if $\boldsymbol{x}_t \neq \boldsymbol{x}_0$ otherwise 1. Thus the computation of $\boldsymbol{p}$ that parameterize $q(\boldsymbol{x}_{t-1}|\boldsymbol{x}_t, \boldsymbol{x}_0)$ breaks down into two cases:

$$
\boldsymbol{p} = \begin{cases} \eta_t\boldsymbol{x}_t + (1-\eta_t)q_{\text{noise}}, & \text{if } \boldsymbol{x}_t = \boldsymbol{x}_0 \\ \lambda_t\boldsymbol{x}_0 + (1-\lambda_t)q_{\text{noise}}(\boldsymbol{x}_t), & \text{if } \boldsymbol{x}_t \neq \boldsymbol{x}_0, \end{cases}
$$

where $\eta_t := 1 - \frac{\beta_t(1-\alpha_{t-1})q_{\text{noise}}(\boldsymbol{u}=\boldsymbol{x}_t)}{\alpha_t+(1-\alpha_t)q_{\text{noise}}(\boldsymbol{u}=\boldsymbol{x}_t)}, \lambda_t := \frac{\alpha_{t-1}-\alpha_t}{1-\alpha_t}$, and $q_{\text{noise}}(\boldsymbol{x}_t) = (1-\beta_t)\boldsymbol{x}_t + \beta_t q_{\text{noise}}$ denotes a noise distribution that interpolates between $\boldsymbol{x}_t$ and $q_{\text{noise}}$.

Since we set $p_{\boldsymbol{\theta}}(\boldsymbol{x}_{t-1}|\boldsymbol{x}_t) = q(\boldsymbol{x}_{t-1}|\boldsymbol{x}_t, f(\boldsymbol{x}_t; \boldsymbol{\theta}))$, the KL divergence between $q(\boldsymbol{x}_{t-1}|\boldsymbol{x}_t, \boldsymbol{x}_0)$ and $p_{\boldsymbol{\theta}}(\boldsymbol{x}_{t-1}|\boldsymbol{x}_t)$ becomes 0 when $\boldsymbol{x}_t = \boldsymbol{x}_0$. In the case of absorbing diffusion, $\boldsymbol{x}_t = q_{\text{noise}} = e_m$ if $\boldsymbol{x}_t \neq \boldsymbol{x}_0$ and $q_{\text{noise}}(\boldsymbol{x}_t) = q_{\text{noise}}$. $\boldsymbol{p}$ has probability $\lambda_t$ on index $\boldsymbol{x}_0$ and $1 - \lambda_t$ on the absorbing state. The model $f(\boldsymbol{x}_t; \boldsymbol{\theta})$ has zero-probability on the absorbing state as it never predicts the mask token. Therefore, $p_{\boldsymbol{\theta}}(\boldsymbol{x}_{t-1}|\boldsymbol{x}_t)$ also has $1 - \lambda_t$ probability on the absorbing state. Putting them together, we derive the KL divergence as:

$$
\begin{aligned}
D_{\text{KL}}[q(\boldsymbol{x}_{t-1}|\boldsymbol{x}_t, \boldsymbol{x}_0)||p_{\boldsymbol{\theta}}(\boldsymbol{x}_{t-1}|\boldsymbol{x}_t)] &= 1_{x_t \neq x_0}[\lambda_t \log \frac{\lambda_t}{f(\boldsymbol{x}_t; \boldsymbol{\theta})_{x_0}} + (1-\lambda_t) \log \frac{1-\lambda_t}{1-\lambda_t}] \\
&= -\lambda_t 1_{x_t \neq x_0} \boldsymbol{x}_0^\top \log f(\boldsymbol{x}_t; \boldsymbol{\theta}) + C,
\end{aligned}
$$

where $1_{x_t \neq x_0}$ is 1 if $x_t \neq x_0$ otherwise 0, and $C$ is a constant. Moreover, given $\alpha_0 = 1$ by definition and therefore $\lambda_0 = 1$, $L_0$ in Eq.(1) can also be written into the final formulation:

$$
L_{\text{vb}} = -\mathbb{E}_{q(\boldsymbol{x}_0)} \sum_{t=1}^T \lambda_t \mathbb{E}_{q(\boldsymbol{x}_t|\boldsymbol{x}_0)} 1_{\boldsymbol{x}_t \neq \boldsymbol{x}_0} \boldsymbol{x}_0^\top \log f(\boldsymbol{x}_t; \boldsymbol{\theta})
$$

For $\mathbf{x}_0$ that represents a sequence of random variables $\mathbf{x}_0 = (\boldsymbol{x}_{0,1}, \ldots, \boldsymbol{x}_{0,N})$, we can add all computed losses for each token, arriving at the final expression for the whole sequence:

$$
L_{\text{vb}} = -\mathbb{E}_{q(\mathbf{x}_0)} \sum_{n=1}^N \sum_{t=1}^T \lambda_t \mathbb{E}_{q(\mathbf{x}_{t,n}|\mathbf{x}_{0,n})} 1_{\mathbf{x}_{t,n} \neq \mathbf{x}_{0,n}} \mathbf{x}_{0,n}^\top \log f(\mathbf{x}_{t,n}; \boldsymbol{\theta}).
$$

For multinomial diffusion, we follow Zheng et al. (2023) to adopt a reparameterized form, which results in the above formulation as well.

## C ADDITIONAL EXPERIMENTS

### C.1 DYNAMIC SEARCH DEPTH IN DIFFUSEARCH

In explicit search algorithms, the search depth is predefined either through an exact parameter as in depth-first search, or a related parameter such as the number of simulations as in MCTS. In DIF-FUSEARCH, the deeper search is achieved by extending the context length of the input. In this

Table 6: Action accuracy when increasing implicit search depth by extending context length. We compare training separate models on $\mathcal{D}$ with different horizon $h$ and a single model on $\mathcal{D}$ with $h = 4$. For the single model, predicting future steps equal to or larger than 4 requires the model's extrapolation ability.

| Future Step | Length | Separate Model | Single Model |
|---|---|---|---|
| 0 | 88 | 23.36 | **34.71** |
| 1 | 168 | 37.35 | **38.11** |
| 2 | 248 | **38.82** | 37.41 |
| 3 | 328 | **41.31** | 36.99 |
| 4 | 408 | **39.34** | 36.78 |
| 5 | 488 | **40.87** | 36.02 |
| 6 | 568 | **41.04** | 34.56 |
| 7 | 648 | **41.69** | 32.72 |

section, we present the results of training a single model for dynamic search depth, compared with separate models in the previous sections. We convert the learned position embedding to RoPE (Su et al., 2024), enabling the utilization of a context length beyond what was encountered during training at inference time. During training, a horizon $h$ is randomly chosen from the interval $[1, 4]$ for each data to expose the model to various input lengths. As shown in Table 6, the single model surpasses the one-step policy with a lookahead of up to 5 future steps, exceeding the training stage's future step of 3. Nonetheless, a diminishing trend emerges as we escalate the search depth, possibly attributed to the constrained training data and limited context extension capability of the current RoPE-based model. We leave more effective context extension beyond training stage for implicit search to future work.

## C.2 SCALING BEHAVIOR WITH MORE DATA

Table 7: Detailed action accuracy with increasing mode size on 10k and 100k games.

| Model Layers | Transformer S-A | DIFFUSEARCH S-ASA |
|---|---|---|
| *10k games (660k records)* | | |
| 1 | 21.92 | 11.97 |
| 2 | 23.28 | 26.61 |
| 4 | 23.05 | 36.49 |
| 8 | 22.10 | 41.31 |
| 16 | 21.55 | 42.87 |
| *100k games (6.6M records)* | | |
| 1 | 29.62 | 11.32 |
| 2 | 35.40 | 31.27 |
| 4 | 36.93 | 42.57 |
| 8 | 36.58 | 48.66 |
| 16 | 35.03 | 51.89 |

**Improved next-action accuracy** In Table 7, we show the comparison of Transformer `S-A` and DIFFUSEARCH `S-ASA` when scaling data and model size. We can see the performance of DIF-FUSEARCH consistently improves with more data and model layers, while that of Transformer `S-A` converges with 2 layers with 10k games and 4 layers with 100k games. Further increasing model size is still useful for DIFFUSEARCH under both data-limited (e.g., 10k games) and relatively data-sufficient (e.g., 100k games) scenarios.

**Improved future accuracy** In Table 8, we show the quality of predicted futures for DIFFUSE-ARCH with horizon $h = 4$. We find when we scale data to 100k games (6.6M records), almost all the future actions are valid (i.e., legal), future states are valid (i.e., the predicted states tokens correctly represent a valid board state), and the action-state transition dynamics are well learned. Moreover, the best action percentage (i.e., action accuracy) also improves greatly compared to that

Table 8: Detailed percentage of future predictions with increasing mode size on 10k and 100k games for DIFFUSEARCH with horizon $h = 4$. Best $a_i$ percentage when future step $i = 0$ is equivalent to action accuracy.

| Future Step | Valid $a_i$ | Best $a_i$ | Valid $s_i$ | Match $a_{i-1}$-$s_i$ |
|---|---|---|---|---|
| *10k games (660k records)* | | | | |
| 0 | 98.40 | 41.31 | 100.00 | - |
| 1 | 79.33 | 20.72 | 97.35 | 37.22 |
| 2 | 50.40 | 4.60 | 53.59 | 6.74 |
| 3 | 50.07 | 3.00 | 51.26 | 3.30 |
| *100k games (6.6M records)* | | | | |
| 0 | 99.85 | 48.66 | 100.00 | - |
| 1 | 99.72 | 32.52 | 99.89 | 99.12 |
| 2 | 99.67 | 19.67 | 99.88 | 99.13 |
| 3 | 99.17 | 13.85 | 99.92 | 93.71 |

in the 10k games setting. This demonstrates the potential of accurate future world modeling through model scaling.

Table 9: Example of training example for each training paradigm. We show horizon $h = 4$ for illustration.

| Paradigm | Input | Output |
|---|---|---|
| S_A | r.bqkbnrpppp.ppp..n.........p.......P........N..PP PP.PPPRNBQKB.RwKQkq-.2.3.. | d2d4 |
| S_V | r.bqkbnrpppp.ppp..n.........p.......P........N..PP PP.PPPRNBQKB.RwKQkq-.2.3.. | WIN[56] |
| SA-V | r.bqkbnrpppp.ppp..n.........p.......P........N..PP PP.PPPRNBQKB.RwKQkq-.2.3.. d2d4 | WIN[61] |
| S-AA | r.bqkbnrpppp.ppp..n.........p.......P........N..PP PP.PPPRNBQKB.RwKQkq-.2.3.. | d2d4 e5d4 f3d4 g8f6 |
| S-ASA | r.bqkbnrpppp.ppp..n.........p.......P........N..PP PP.PPPRNBQKB.RwKQkq-.2.3.. | d2d4 r.bqkbnrpppp.ppp..n.........p......PP.......N..PPP..PPPRNBQKB.RbKQkq-.0.3.. e5d4 r.bqkbnrpppp.ppp..n...............pP........N..PPP..PPPRNBQKB.RwKQkq-.0.4.. f3d4 r.bqkbnrpppp.ppp..n................NP...........PPP..PPPRNBQKB.RbKQkq-.0.4.. g8f6 |
| S-ASS | r.bqkbnrpppp.ppp..n.........p.......P........N..PP PP.PPPRNBQKB.RwKQkq-.2.3.. | d2d4 r.bqkbnrpppp.ppp..n.........p......PP........N..PPP..PPPRNBQKB.RbKQkq-.0.3.. r.bqkbnrpppp.ppp..n...............pP........N..PPP..PPPRNBQKB.RwKQkq-.0.4.. r.bqkbnrpppp.ppp..n................NP...........PPP..PPPRNBQKB.RbKQkq-.0.4.. |
| S-AVAV | r.bqkbnrpppp.ppp..n.........p.......P........N..PP PP.PPPRNBQKB.RwKQkq-.2.3.. | d2d4 WIN[61] e5d4 WIN[69] f3d4 WIN[60] g8f6 WIN[68] |
| S-AVSAV | r.bqkbnrpppp.ppp..n.........p.......P........N..PP PP.PPPRNBQKB.RwKQkq-.2.3.. | d2d4 WIN[61] r.bqkbnrpppp.ppp..n.........p......PP.......N..PPP..PPPRNBQKB.RbKQkq-.0.3.. e5d4 WIN[69] r.bqkbnrpppp.ppp..n...............pP........N..PPP..PPPRNBQKB.RwKQkq-.0.4.. f3d4 WIN[60] r.bqkbnrpppp.ppp..n................NP...........PPP..PPPRNBQKB.RbKQkq-.0.4.. g8f6 WIN[68] |

### C.3 EXAMPLE OF TRAINING INSTANCE

We show an example of each training paradigm in Table 9.

### C.4 ADDITIONAL CASES

In Figure 5, we provide the predictions of Transformer (S-A) and DIFFUSEARCH on more challenging puzzles. We also show the prediction of all models in Figure 6, where all models are trained on 10k games and 100 MCTS simulations are used for Transformer with MCTS.

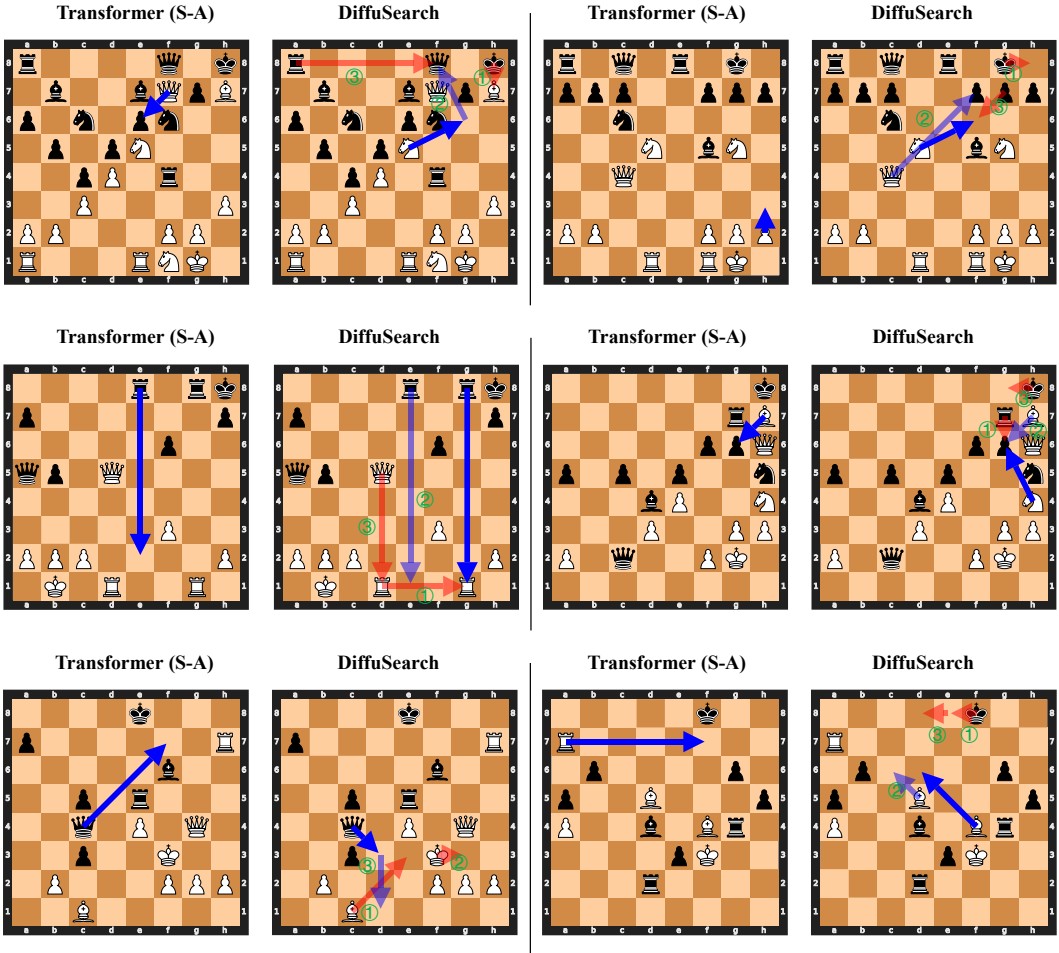

Figure 5: Additional prediction cases on challenging puzzles.

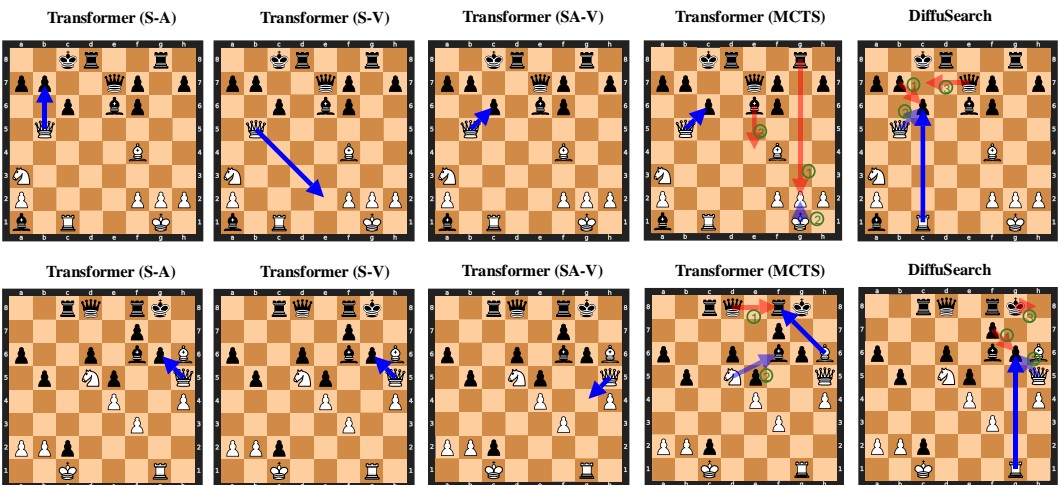

Figure 6: Additional cases on challenging puzzles compared with all baselines.

