# OpenReview forum: "Implicit Search via Discrete Diffusion: A Study on Chess"
_ICLR.cc/2025/Conference — ICLR 2025 Poster_

### Official Review · Reviewer_LQsg · 2024-11-03

**Soundness:** 4
**Presentation:** 3
**Contribution:** 3
**Rating:** 6
**Confidence:** 3

**Summary:**

DiffuSearch is a novel approach to enhancing the planning abilities of Large Language Models (LLMs) without relying on explicit search techniques like Monte Carlo Tree Search (MCTS). Developed in response to the limitations of current next-token prediction models in long-term planning, DiffuSearch uses discrete diffusion modeling to perform implicit search by predicting and utilizing future states. The method is implemented and tested on the game of chess, where explicit search has traditionally been essential. In extensive experiments, DiffuSearch outperforms both searchless and explicit search-enhanced policies, demonstrating a 19.2% improvement over one-step policies and a 14% improvement over MCTS-enhanced policies in action accuracy. Additionally, it shows a 30% enhancement in puzzle-solving abilities and a significant 540 Elo increase in game-playing strength compared to explicit search methods. These results suggest that DiffuSearch’s approach of internalizing a world model within the policy, without intermediate components, could be a promising alternative to traditional explicit search techniques in AI problem-solving.

**Strengths:**

1. Superior performance: DiffuSearch significantly outperforms baseline models, as evidenced in Table 2. It demonstrates a substantial improvement over both searchless and explicit search-enhanced policies, showing the potential of using diffusion as a model for search in chess.
2. Novel methodology: DiffuSearch incorporates future states and actions through discrete diffusion modeling, enabling it to leverage future information for improved next action prediction without relying on explicit search. This approach offers a new perspective on implicit search in AI.
3. Sample efficiency: Despite using 20 times fewer data records than some baseline models (e.g., SA-V), DiffuSearch demonstrates superior performance with approximately 10% higher action accuracy.

**Weaknesses:**

1. Declining prediction accuracy: As shown in Figure 2 (left), the accuracy of predicted future states and actions declines significantly for steps further into the future. For a strong world model in chess, the lookahead should ideally be accurate for about 7 steps, similar to top engines like Stockfish.
2. Training complexity: The paper doesn't provide a clear comparison of the computational requirements (e.g., FLOPs) for training DiffuSearch versus traditional transformer models. This makes it difficult to assess the scalability and efficiency of the diffusion process compared to other approaches.
3. Limited comparison to state-of-the-art: The paper doesn't compare DiffuSearch to more recent advancements in chess AI [1] that achieved a 2299 Elo rating using transformers. This omission makes it challenging to contextualize DiffuSearch's performance within the current state-of-the-art in chess AI.
[1] Ruoss, A., Delétang, G., Medapati, S., Grau-Moya, J., Wenliang, L. K., Catt, E., ... & Genewein, T. (2024). Grandmaster-level chess without search. arXiv preprint arXiv:2402.04494.

**Questions:**

Line 060: It is unclear if the Ha & Schmidhuber citation is correct or necessary here. Work has existed prior to this paper[2] and there are many more works that use it. This is a fundamental concept and probably does not need a citation unless it is citing a textbook. It should be removed.

Is the performance conditioned on the amount of train time and test time compute? What is the trade-off in FLOPs at train time versus test time?

Line 083: Code link is broken. Make sure to correct this for publication.


[2] Triggs, B., and Cameron, S. (1991). ““The Oxford robot world model,”,” in Expert systems and robotics (Springer Berlin Heidelberg), 275–284.

---

> ### Author Response · Authors · 2024-11-21
>
> We sincerely thank Reviewer LQsg for your review and are grateful for the time you spent on our submission. We are also glad you think our methodology is novel and offers a new perspective on implicit search in AI. Below we would like to give detailed responses to each of your comments.
>
> **W1: Declining prediction accuracy: As shown in Figure 2 (left), the accuracy of predicted future states and actions declines significantly for steps further into the future. For a strong world model in chess, the lookahead should ideally be accurate for about 7 steps, similar to top engines like Stockfish.**
>
> Thank you for your insightful comments. We believe that one significant factor contributing to the diminishing effectiveness of lookahead as the future steps increase is related to the volume of data.  In Figure 2, we evaluated a model trained on 10k games (660k records), and we further discovered that scaling the data to 100k games (6.6M records) in the Table below (Table 8 in the paper) significantly improved performance. In the world model construction (Valid $s_i$ and Match  $a_{i-1} - s_i$), the model achieves over 90% accuracy within 3 steps, while the 10k data trained model declined to around 50% in predicting valid future states. Therefore, we are optimistic that further scaling of the data could also enhance the world construction accuracy for the 7 steps.
>
> Additionally, our goal is not to build a highly competitive chess engine, such as the ones that rely on extensive training like Lc0. Instead, we hope to propose a different paradigm for solving complex problems, beyond the one-step policy with explicit search, which may provide insights into building LLMs with enhanced reasoning and planning capabilities.
>
> | Future Step | Valid $a_i$ | Best $a_i$ | Valid $s_i$ | $a_{i-1}$-$s_{i}$ match |
> |---|---|---|---|---|
> | 10k games (660k records) |  |  |  |  |
> | 0 | 98.40 | 41.31 | 100.00 | - |
> | 1 | 79.33 | 20.72 | 97.35 | 37.22 |
> | 2 | 50.40 | 4.60 | 53.59 | 6.74 |
> | 3 | 50.07 | 3.00 | 51.26 | 3.30 |
> | 100k (6.6M records) |  |  |  |  |
> | 0 | 99.85 | 48.66 | 100.00 | - |
> | 1 | 99.72 | 32.52 | 99.89 | 99.12 |
> | 2 | 99.67 | 19.67 | 99.88 | 99.13 |
> | 3 | 99.17 | 13.85 | 99.92 | 93.71 |
>
> **W2: Training complexity: The paper doesn't provide a clear comparison of the computational requirements (e.g., FLOPs) for training DiffuSearch versus traditional transformer models. This makes it difficult to assess the scalability and efficiency of the diffusion process compared to other approaches.**
>
> Thank you for your feedback. In Figure 2 (middle) and Appendix C.2, we have demonstrated that DiffuSearch exhibits scalability as the model and data size increase. Below, we provide a comparison based on FLOPS for reference. Overall, we find DiffuSearch outperforms the Transformer at equivalent FLOPS, and both the Transformer and DiffuSearch show that increasing FLOPS with more data or larger model sizes yields improvements. However, under the same amount of data, Transformers are more prone to saturation, leading to diminishing returns when increasing model size. In contrast, DiffuSearch, due to its more challenging objective, demonstrates more improvements when the model size increases. Additionally, both models show substantial gains when scaling data size.
>
> | FLOPS | Transformer | DiffuSearch |
> |---|---|---|
> | 3.7e17 | 27.39 | 32.17 |
> | 1.8e18 (5x model size) | 28.01 | 37.37 |
> | 1.8e18 (5x data) | 34.78 | 42.52 |
>
>
> **W3: Limited comparison to state-of-the-art: The paper doesn't compare DiffuSearch to more recent advancements in chess AI [1] that achieved a 2299 Elo rating using transformers. This omission makes it challenging to contextualize DiffuSearch's performance within the current state-of-the-art in chess AI.**
>
> Thanks for bringing up this point. On the one hand, [1] annotated 10M games from Stockfish and trained on 128 TPUs, which is unaffordable for us. Due to the resource constraint, we consider a more affordable and controlled comparison between them throughout the experiments, where the Transformer S-A, Transformer S_V, and Transformer SA-V baselines are their models but trained with the same data size with DiffuSearch. We can see that DiffuSearch has a performance advantage compared to these baselines.
>
> On the other hand, similar to W1, our goal is not to build a highly competitive chess engine. Instead, we hope to propose a different paradigm for solving complex problems, beyond the one-step policy [1] and one-step policy with explicit search, which may provide insights into building LLMs with enhanced reasoning and planning capabilities.

---

> > ### Author Response · Authors · 2024-11-21
> >
> > **Q1: Line 060: It is unclear if the Ha & Schmidhuber citation is correct or necessary here. Work has existed prior to this paper[2] and there are many more works that use it. This is a fundamental concept and probably does not need a citation unless it is citing a textbook. It should be removed.**
> >
> > Thank you for your suggestion. We have made revisions in the latest version.
> >
> > **Q2:  Is the performance conditioned on the amount of train time and test time compute? What is the trade-off in FLOPs at train time versus test time?**
> >
> > Yes. We show in the experiments and W2 that increasing training data and model size greatly improves performance.  Because the size of training data is much larger than that of inference data, the direct comparison of FLOPS shows that there are more FLOPs during training time (e.g., 3.7e17) compared to inference time (e.g., 2.9e14). We further find the performance with a 5x increase in FLOPs during inference time (e.g., increasing diffusion timesteps) is lower than that with a 5x increase in FLOPs during training time (i.e., increasing data size). However, if we strictly compare FLOPs, a 5x increase in inference time FLOPs only leads to a negligible performance improvement (from 32.17 to 32.21) when translated in training time, which is only about 1.003 times training FLOPs compared to the original model.
> >
> > | Setting | Train FLOPS | Infer FLOPS | Acc |
> > |---|---|---|---|
> > | Base  | 3.7e17 | 2.9e14 | 32.17 |
> > | 5x FLOPs on training time | 1.8e18 | 2.9e14 | 42.52 |
> > | 1.003x FLOPs on training time | 3.71e17 | 2.9e14 | 32.21 |
> > | 5x FLOPs on inference time | 3.7e17 | 1.5e15 | 33.21 |
> >
> > **Q3: Line 083: Code link is broken. Make sure to correct this for publication.**
> >
> > Thanks for the comment. We have attached the code as supplementary material for your interest and will modify Line 083 to an authorized link for publication.
> >
> > Hope our response could address your questions!

---

> > > ### Author Response · Authors · 2024-11-27
> > >
> > > Dear Reviewer LQsg,
> > >
> > > Thank you for your valuable time to review our work and for your constructive feedback. As the author-reviewer discussion period is coming to a close, we wonder if you could kindly take a look at both the revision and our response to your comments. We would appreciate it if you could consider adjusting the score based on our responses and the other review comments. If you have any further questions, we are happy to discuss them!
> > >
> > > Best regards,
> > >
> > > Authors

---

> > > > ### Author Response · Authors · 2024-12-03
> > > >
> > > > Dear Reviewer LQsg,
> > > >
> > > > Thank you for your valuable time to review our work and for your constructive feedback. As the author-reviewer discussion period is coming to a close, we wonder if you could kindly take a look at both the revision and our response to your comments. We would appreciate it if you could consider adjusting the score based on our responses and the other review comments. If you have any further questions, we are happy to discuss them!
> > > >
> > > > Best regards,
> > > >
> > > > Authors

---

> > > > > ### Comment · Reviewer_LQsg · 2024-12-03
> > > > >
> > > > > I have reviewed the rebuttals and affirm that I would need higher Elo performance to increase my score further. However, I think the evaluation is sufficient for a current publication version and urge other reviewers to raise their scores.

---

> > > > > > ### Author Response · Authors · 2024-12-03
> > > > > >
> > > > > > Thanks for your timely feedback and we truly appreciate your recognition. Since our goal is not to achieve SOTA Elo performance but to validate the feasibility of the new paradigm of "diffusion as implicit search", by rigorously comparing the performance of DiffuSearch with other baselines (e.g., searchless and explicit-search enhanced one-step policy) and demonstrating a greater advantage in Elo and other metrics, we also believe we have provided sufficient evidence to respond to our research question. We leave the extensive scaling of data and model size to reach a SOTA performance level on chess to future work due to current resource constraints.

---

### Official Review · Reviewer_XeCm · 2024-11-03

**Soundness:** 3
**Presentation:** 2
**Contribution:** 3
**Rating:** 5
**Confidence:** 3

**Summary:**

This paper trains a transformer to imitate actions from a chess engine, while using discrete diffusion modeling to incentivize a form of implicit search during action selection. The diffusion modeling distills forward and backward prediction into the policy network, whereas the baseline transformer with MCTS involves only forward prediction. The paper includes several comparisons and ablations, and claims that diffusion modeling improves performance.

**Strengths:**

The paper investigates whether diffusion modeling can be helpful for emulating search using a feedforward network. This is an interesting question, as transformers have generally struggled thus far to solve problems requiring search. Normally, the solution is to add explicit search in the form of MCTS using the outputs of the transformer. This paper tries to improve the policy network instead, and indeed provides some evidence that transformers can simulate search using search with a single forward pass.

**Weaknesses:**

1. I wish it were more clear what exactly the paper is arguing. The paper seems to provide evidence for the following claim: "If we must use transformers in tasks that require search, it is more efficient and effective to train them via diffusion to do implicit search than it is to add explicit search via MCTS." However, it seems as though the paper is instead arguing that implicit search is better than explicit search. There doesn't seem to be evidence for this, as the method still relies on a dataset that comes from a Stockfish oracle that uses explicit search (and which vastly outperforms the diffusion model). If the paper is arguing for the former, weaker claim, then I think it does an fine job of this, provided the intro/discussion/etc. are updated to make this more clear.

2. I would have liked to see much more detail on the MCTS baseline. I couldn't tell whether it uses a perfect world model or a learned one, and if the world model is learned, I couldn't find information on how it is trained. These details are crucial for understanding exactly what method diffusion is improving on. How does the paper combine the policy model with MCTS to do search for each different "future paradigm"? Furthermore, the future paradigms in section 3.1 include s-arsar, which seems to be different from s-avsav, and while these names provide an intuition about what's going on, it's still unclear precisely what either of these mean.

3. The text is often rather imprecise, and in some places even a bit misleading. For instance:
    1. AlphaGo pre-dates the transformer, but the abstract makes it seem like people were working on LLMs prior to AlphaGo.
    2. Heuristic search existed long before the deep learning revolution, but the intro makes it sound like NNs introduced the idea of more efficient search.
    3. The intro (line 044) references three papers, all of which discuss model-based RL and the compounding error problem, but presents them as evidence that such errors occur due to recursive invocation of the *value* function, rather than the world model.
    4. The problem setting defines the value function with respect to a single policy, despite the fact that the players generally have different policies.
    5. In Sec 4.5 (line 288), S-AVAV does not lead to a significant performance improvement, though S-ASS does.
    6. It is unclear what the names in Table 4 actually mean. The process is described rather vaugely in the adjacent paragraph.
    7. The scaling trends in lines 411-418 are presented as scaling "laws", but it is far from clear that these results will have that level of robustness beyond that single experiment in this paper.
    8. Line 429 suggests that the method correctly values "the long-term positional benefits of opening lines for its rooks." However, the model is playing as white, has only one rook, and sacrifices it in the first turn, preventing any such long-term benefits.
    9. Section 5.2 (line 468), suggests that diffusion models might be "a way to substitute for conventional handcrafted search algorithms," but offers no evidence of this claim, since the oracle that provides the training data for the model uses exactly that sort of conventional handcrafted search algorithm.

**Questions:**

1. Can you provide more detail on precisely how MCTS is being combined with the policy network?

2. The Best $a_i$ and Match $a_{i-1}-s_i$ lines in Fig. 2 (left) are not discussed in the text. It seems like the model is terribly inaccurate for the actions it actually takes. How does it do so well?

3. In Table 8 (Appendix B), if $s'$ matches $f(s,a)$ with 99% accuracy, why does Best $a_i$ accuracy drop by ~1/3 after each step?

---

> ### Author Response · Authors · 2024-11-21
>
> We sincerely thank Reviewer XeCm for your detailed review and are grateful for the time you spent on our submission. We are also glad you think our research question is interesting. Below we would like to give detailed responses to each of your comments.
>
> **W1: About the paper claim.**
>
> Thank you for your insightful suggestion. The position of this work exactly aligns with the weaker claim you mentioned: given the popularity of enhancing one-step policy (e.g., LLMs) inference with explicit search, we want to explore alternative solutions to enhance their planning abilities. We find implicit search via discrete diffusion indeed has the potential to compete with one-step policy with explicit search through controlled experiments. Sometimes we use "explicit search" to directly refer to the “one-step policy with explicit search” for brevity, which may confuse as you mentioned. In the updated version, we have made proper corrections.
>
> **W2: I would have liked to see much more detail on the MCTS baseline. I couldn't tell whether it uses a perfect world model or a learned one, and if the world model is learned, I couldn't find information on how it is trained. These details are crucial for understanding exactly what method diffusion is improving on.**
>
> Thanks for your suggestion. To combine MCTS and the one-step policy, we follow the approach of AlphaZero, which uses a perfect world model to perform action-state transitions. We have added more detail about the MCTS-enhanced policy baseline in the Appendix B of the latest version.
>
> > How does the paper combine the policy model with MCTS to do search for each different "future paradigm"?
> >
>
> Thank you for your question. The term "different future paradigm" specifically refers to DiffuSearch; we did not adjust the future paradigm for the policy model with MCTS. The employed policy model with MCTS is the standard approach following AlphaZero. We have added more detail about the MCTS baseline in the latest version.
>
> > Furthermore, the future paradigms in section 3.1 include s-arsar, which seems to be different from s-avsav, and while these names provide an intuition about what's going on, it's still unclear precisely what either of these mean.
> >
>
> Thanks for noting this typo. s-arsar means the same as s-avsav and we will use s-avsav for consistency. We have fixed the typo and added specific examples of each paradigm in the Appendix C.3 to make it clearer.
>
> **W3: The text is often rather imprecise, and in some places even a bit misleading.**
>
> Thanks for your detailed comments and pointing out some potential confusion. We will explain each point below and refine the corresponding descriptions in the updated version.
>
> > 1. AlphaGo pre-dates the transformer, but the abstract makes it seem like people were working on LLMs prior to AlphaGo.
> >
> >2. Heuristic search existed long before the deep learning revolution, but the intro makes it sound like NNs introduced the idea of more efficient search.
> >
>
> Thank you for pointing out this potential confusion. We have made updates to avoid any ambiguity.
>
>
> > 3. The intro (line 044) references three papers, all of which discuss model-based RL and the compounding error problem, but presents them as evidence that such errors occur due to recursive invocation of the *value* function, rather than the world model.
> >
>
> Thank you for your insightful feedback. We would like to clarify that the one-step policy with MCTS is based on a learned value model to guide search at each step and a perfect world model to perform action-state transitions. This paradigm inherently leads to a recursive invocation of the value function, which can result in a compounding error problem.
>
> > 4. The problem setting defines the value function with respect to a single policy, despite the fact that the players generally have different policies.
> >
>
> Thank you for your feedback. The modeling of DiffuSearch is independent of whether it involves multiple policies. Since we are focusing on chess, we used it as a research example and adopted a simplified definition. However, extending this to a multiple-policy scenario is certainly possible and represents an interesting and valuable direction.
>
> > 5. In Sec 4.5 (line 288), S-AVAV does not lead to a significant performance improvement, though S-ASS does.
> >
>
> Thank you for your comments. In Line 288 (i.e., Line 293 in the updated version), we focus on comparing different paradigms under the same DiffuSearch modeling framework. We illustrate the significant improvements achieved by incorporating the future state "S" into the model. For instance, we observe enhancements from S-AA (15.07) to S-ASA (41.31), as well as from S-AVAV (17.63) to S-AVSAV (40.69). If I understand correctly, the reviewer may interpreted from another angle, i.e., the comparison between Transformer and DiffuSearch within the same paradigm, which is also correct. We have added clarifications in the text to prevent any ambiguity.

---

> > ### Author Response · Authors · 2024-11-21
> >
> > > 6. It is unclear what the names in Table 4 actually mean. The process is described rather vaugely in the adjacent paragraph.
> > >
> >
> > Thank you for your feedback. Here we provide a more detailed description of these variants. Denote a sequence of future horizon 2 as $[s_1=f(s_0,a_0),a_1=g(s_1),s_2=f(s_1,a_1),a_2=g(s_2)]$, where f is a world dynamic function and g is a policy function. $s_0$ is the current state and $a_0$ is the move suggested by Stockfish. “Without future world” refers to the S-A baseline which directly learns to predict $a_0$ from $s_0$. “Corrupted world” refers to the use of a random $f$ (i.e., output random state) and a random $g$ (i.e., output random action). “Random world” refers to the use of a random g  but an oracle $f$ (i.e., perfect world dynamic). “Oracle world” refers to the use of both oracle $f$ and $g$ (i.e., Stockfish). We have added details in the updated version.
> >
> > > 7. The scaling trends in lines 411-418 are presented as scaling "laws", but it is far from clear that these results will have that level of robustness beyond that single experiment in this paper.
> > >
> >
> > Thanks for the feedback. We have observed that as the model layer increases, Transformers and DiffuSearch exhibit different characteristics in Figure 2 (middle), and scaling data size also leads to consistent performance boosts for both Transformers and DiffuSearch. Developing a more detailed scaling law with larger models and more data will certainly be a valuable future work once we have the necessary computational resources.
> >
> > > 8. Line 429 suggests that the method correctly values "the long-term positional benefits of opening lines for its rooks." However, the model is playing as white, has only one rook, and sacrifices it in the first turn, preventing any such long-term benefits.
> > >
> >
> > Thanks for noting this typo. In the latest version, we have clarified that the model sacrifices rooks to open lines for the queen.
> >
> > > 9. Section 5.2 (line 468), suggests that diffusion models might be "a way to substitute for conventional handcrafted search algorithms," but offers no evidence of this claim, since the oracle that provides the training data for the model uses exactly that sort of conventional handcrafted search algorithm.
> > >
> >
> > Thank you for raising this point. We have rephrased it to: "Furthermore, we focus on exploring diffusion models for implicit search as an alternative to the one-step policy with explicit search to deal with complex tasks that require search.".
> >
> > **Q1: Can you provide more detail on precisely how MCTS is being combined with the policy network?**
> >
> > Thank you for your question. This baseline is fully aligned with the approach used in AlphaZero. Here, we briefly describe how MCTS is combined with the policy and refer reviewers to Appendix B for a detailed description.
> >
> > One-step policy directly predicts the next action, while MCTS is integrated to construct a search tree that simulates the future to enhance the evaluation of potential next actions. MCTS consists of four essential phases:
> >
> > 1. **Selection**: The algorithm begins at the root node and traverses the tree, selecting child nodes based on strategies such as Upper Confidence Bound for Trees (UCT) to maximize the exploration of promising paths.
> > 2. **Expansion and evaluation**: Upon reaching a leaf node, if it does not represent a terminal state (i.e., the end of the game), one or more new child nodes are expanded and evaluated by the policy and value model.
> > 3. **Backup**: The evaluation result is propagated back up the tree, updating the statistical information (e.g., visit counts and action-value) for each node along the path.
> > 4. **Play:** After iteratively cycling through the above phases, a move is selected to play in the root position at the end of the search based on the statistical information.
> >
> > **Q2: The Best $a_i$ and Match $a_{i-1}-s_i$ lines in Fig. 2 (left) are not discussed in the text. It seems like the model is terribly inaccurate for the actions it actually takes. How does it do so well?**
> >
> > Thank you for your insightful question. The observed decline in the performance of the Best $a_i$ and Match $a_{i-1}-s_i$ indicates that predicting future steps becomes increasingly challenging given initial state $s_0$. Note that only predicted $a_0$ is the action that the model actually takes and other $a_i$ and $s_i$ when $i>0$ are “imagined” by the model to improve the prediction of $a_0$. We've optimized Figure 1 to make it clearer.
> >
> > **Q3: In Table 8 (Appendix B), if s′ matches f(s,a) with 99% accuracy, why does Best ai accuracy drop by ~1/3 after each step?**
> >
> > Thank you for your great question. Similar as in Q2, since only the predicted $a_0$ is the action that the model actually takes and the latter $a_i$ and $s_i$ are “imagined” by the model for improving the prediction of $a_0$, the difficulty of predicting further future steps will increase given only the initial state $s_0$.
> >
> > Hope our response could address your questions!

---

> > ### Comment · Reviewer_XeCm · 2024-11-25
> >
> > Thanks for the clarifications. I'll just follow up on one point here.
> >
> > 3.
> >
> > > the one-step policy with MCTS is based on a learned value model to guide search at each step and a perfect world model to perform action-state transitions. This paradigm inherently leads to a recursive invocation of the value function, which can result in a compounding error problem.
> >
> > I'm not sure what you mean by compounding error problem then. Normally the compounding error problem involves taking the outputs of a function (which have some small prediction error) and feeding them back in as _inputs_ to the same function, with prediction errors accumulating with each successive function application. The input of a value function is a state(-action) but the output is a utility, so there's no way to feed outputs back into the model. If you're talking about _bootstrapping_, that's another matter entirely, and these are the wrong references.

---

> ### Comment · Reviewer_XeCm · 2024-11-25
>
> 6. I'm not sure why you call a perfect world model with random actions "random world". That still feels confusing to me. Why not just call them "No world model", "Random world+policy", "Random policy", and "Stockfish"---or something similar?
>
> 7. I agree that would be a valuable direction. But if what you have isn't actually a scaling "law", I would suggest you don't call it that.
>
> Q2.
>
> > The observed decline in the performance of [...].
>
> I'm not talking about the decline in performance, I'm talking about the initial performance. These curves only _start_ at ~35-40% accuracy. This says even at the very first step, the model isn't able to accurately predict the best action or the resulting state.

---

> > ### Author Response · Authors · 2024-11-25
> >
> > **Q: I'm not sure what you mean by compounding error problem then. Normally the compounding error problem involves taking the outputs of a function (which have some small prediction error) and feeding them back in as inputs to the same function, with prediction errors accumulating with each successive function application. The input of a value function is a state(-action) but the output is a utility, so there's no way to feed outputs back into the model. If you're talking about bootstrapping, that's another matter entirely, and these are the wrong references.**
> >
> > Thank you for your feedback. We originally intended to express that the search-based one-step policy requires multiple calls to the value model in the search process, and when the value model is inaccurate, it can mislead the process, leading to cumulative errors. For example, in MCTS, the next action is selected based on the estimated Q value from previous explorations (Line 934). If the value model is not accurate, it will impact the Q value and the choice of the next action, as well as the subsequent state. Despite the presence of an exploration mechanism such as PUCT, this error will still influence the selection of future actions and states during the search process.
> >
> > In summary, this is more like “error accumulation in the sequential search process due to the inaccurate value model”. This bears some high-level similarity to the compounding error in the one-step world model where error accumulates in the sequential generation process of the world model. However, we acknowledge that it is not a strict match, and we have made corrections in the updated version (Line 41-44) to make it more rigorous.
> >
> > **Q: I'm not sure why you call a perfect world model with random actions "random world". That still feels confusing to me. Why not just call them "No world model", "Random world+policy", "Random policy", and "Stockfish"---or something similar?**
> >
> > Thank you for your suggestion. We have changed the words in Table 4 in the updated version to make it clearer.
> >
> > **Q: I agree that would be a valuable direction. But if what you have isn't actually a scaling "law", I would suggest you don't call it that.**
> >
> > Thank you for your suggestion. We choose to delete the term "law” in the updated version to make it more accurate.
> >
> > **Q: I'm not talking about the decline in performance, I'm talking about the initial performance. These curves only start at ~35-40% accuracy. This says even at the very first step, the model isn't able to accurately predict the best action or the resulting state.**
> >
> > Thank you for your feedback. Regarding the performance of the first action, this metric is actually equivalent to Action Accuracy. We found that DiffuSearch outperforms other baselines, as shown in Table 2, and the performance improves further with more training data.
> > |Model  | Action Acc |
> > |:---:|:---:|
> > | 10k games |  |
> > | Transformer (S-A) | 22.10 |
> > | Transformer  (S-V) | 21.45 |
> > | Transformer  (SA-V) | 31.50 |
> > | Transformer (100 MCTS simulations) | 27.34 |
> > | DiffuSearch (Ours) | 41.31 |
> > | 100k games |  |
> > | Transformer (S-A) | 36.58 |
> > | Transformer  (S-V) | 28.89 |
> > | Transformer  (SA-V) | 39.76 |
> > | Transformer (100 MCTS simulations) | 38.05 |
> > | DiffuSearch (Ours) | 48.66 |
> >
> > As for the performance of the first state, we discovered that scaling can significantly enhance performance. As demonstrated in Table 8 and Lines 1016-1022, using ten times the previous amount of data achieved 99% accuracy in predicting the first state. We have added more descriptions about this in the updated version (Line 368-371).
> > | Future Step | Valid $a_i$ | Best $a_i$ | Valid $s_i$ | $a_{i-1}$-$s_{i}$ match |
> > |---|---|---|---|---|
> > | 10k games (660k records) |  |  |  |  |
> > | 0 | 98.40 | 41.31 | 100.00 | - |
> > | 1 | 79.33 | 20.72 | 97.35 | 37.22 |
> > | 2 | 50.40 | 4.60 | 53.59 | 6.74 |
> > | 3 | 50.07 | 3.00 | 51.26 | 3.30 |
> > | 100k (6.6M records) |  |  |  |  |
> > | 0 | 99.85 | 48.66 | 100.00 | - |
> > | 1 | 99.72 | 32.52 | 99.89 | 99.12 |
> > | 2 | 99.67 | 19.67 | 99.88 | 99.13 |
> > | 3 | 99.17 | 13.85 | 99.92 | 93.71 |
> >
> > Thank you again for your valuable feedback on improving our work. We hope our response could address your questions, and we are happy to address any further concerns or queries.

---

> > > ### Comment · Reviewer_XeCm · 2024-11-26
> > >
> > > Thank you for the response. This does clarify things a bit. I guess I've been looking at this from the perspective of whether the predictions are accurate enough to trust them, but when you compare against all the other models' action accuracy numbers, it helps put things in perspective. None of these models is what I would call "good" at predicting actions, but I suppose DiffuSearch is at least better than the baselines.
> > >
> > > I am glad you removed the bit about compounding error. I'm not trying to pick nits here, but "recursive invocation [of the value model]" should probably be "repeated invocation", as it is not being invoked recursively, for the same reasons I outlined above. I do think it's important to really nail the high-level story in the introduction, or readers (like me) will be confused about what exactly you're trying to do and why.

---

> > > > ### Author Response · Authors · 2024-11-26
> > > >
> > > > Thanks for your valuable feedback. We have changed the term "recursive" to "repeated" to make it more accurate. Besides, to better provide a high-level story of our work in the introduction, we have moved Figure 1 into that section and added relevant descriptions (Lines 72-74) to help the understanding of the objective and method part in the introduction (Lines 50-74).
> > > > Additionally, we have enhanced the description of the objective in the caption of Figure 1 (...while discrete diffusion implicitly gathers future information during future imagination **to improve the next action**) and the research question (Lines 47-48, Can the policy model predict and utilize the future by itself **to improve the next action
> > > > prediction** without relying on explicit search during inference?) to emphasize the objective. With these changes, we expect the high-level story can be better conveyed in the introduction.
> > > >
> > > > Thank you again for your valuable feedback on improving our work. We hope our response could address your questions, and we are happy to address any further concerns or queries.

---

> > > > > ### Author Response · Authors · 2024-12-03
> > > > >
> > > > > Dear Reviewer XeCm,
> > > > >
> > > > > Thank you for your valuable time to review our work and for your constructive feedback. As the author-reviewer discussion period is coming to a close, we wonder if you could kindly take a look at both the revision and our response to your comments. We would appreciate it if you could consider adjusting the score based on our responses and the other review comments. If you have any further questions, we are happy to discuss them!
> > > > >
> > > > > Best regards,
> > > > >
> > > > > Authors

---

### Official Review · Reviewer_KzTF · 2024-11-04

**Soundness:** 2
**Presentation:** 3
**Contribution:** 3
**Rating:** 6
**Confidence:** 3

**Summary:**

This work aims to explore the possibility of planning using LLM without explicit search (e.g., MCTS), i.e., employing implicit search. Specifically, this paper proposes a method named DiffuSearch that looks into the future world by diffusion modeling. DiffuSearch considers the bidirectional self-attention  architecture and the multi-step diffusion generative process. This work focuses on a study of a specific board game--chess. The numerical experiments demonstrate the efficacy of the proposed DiffuSearch approach.

**Strengths:**

Overall, the idea of this work is novel and intuitive.

The paper is well-written and easy-to-follow.

The technical contents like the theorems as well as proofs are well-organized and sound.

The experimental setup is detailed and clear. The empirical results substantiate that DiffuSearch outperforms the existing baselines.

The demonstration plots like Figures 4 and 5 are very clear and intuitive.

**Weaknesses:**

Line 83. The link of source code is invalid. I hope to see the code during the rebuttal. The authors may consider submitting via a zip file or providing a valid link to an anonymous repo.


I would expect more detailed explanation of Figure 1 in both the main texts and the caption of Figure 1. As the comparison between the explicit and implicit searches is the main idea of this work. For example, the authors could explain the difference of explicit and implicit searches by describing Figure 1 more carefully. The discussion of the structures in Figure 1 is not enough. I felt that the difference between the explicit and implicit that authors mentioned in the texts is not well-connected to Figure 1.

The authors should provide a brief explanation of each input parameter and its role in Algorithm 1. For instance, what is \lambda_t? And a curiosity question: can you use other random sampling methods to draw t in line 169? Like Gaussian? Will different sampling methods affect the algorithm's performance?


In Figures 3(a) and (b), why DiffuSearch only runs around 7 depths? Are there any technical limitations that might prevent running DiffuSearch at greater depths? If it is feasible, I would like to see if it converges when you run the same depth as the baseline method, i.e., depth = 24 or 25.


I noticed that the demonstration plots like Figures 4 and 5 are limited to the comparison between DiffuSearch and Transformer (S-A). Can the authors explain the possible reasons? What about DiffuSearch against Transformer (S-V), against Transformer (SA-V), and Transformer (100 MCTS simulations)? If it is feasible, the authors may show at least one scenario for all of these three baseline comparisons. If not feasible, can the authors provide some explanations/reasons?

**Questions:**

Please refer to the "Weaknesses" section.

---

> ### Author Response · Authors · 2024-11-21
>
> We sincerely thank Reviewer KzTF for your review and are grateful for the time you spent on our submission. We’re pleased you find our paper novel and intuitive. Below, we provide a point-by-point rebuttal to clarify your concerns.
>
> **W1: Line 83. The link to the source code is invalid. I hope to see the code during the rebuttal.**
>
> Thanks for your interest. We have attached the code above.
>
> **W2: I would expect more detailed explanation of Figure 1 in both the main texts and the caption of Figure 1.**
>
> Thanks for the suggestion. MCTS explicitly performs action selection, state evaluation, and value backup in an iterative manner before determining the final action to take, while discrete diffusion implicitly gathers future information during the process of future imagination. We have optimized Figure 1, provided additional explanations in the caption, and placed the more detailed content about explicit search via MCTS in the Appendix A.
>
> **W3: The authors should provide a brief explanation of each input parameter and its role in Algorithm 1. For instance, what is $\lambda_t$? And a curiosity question: can you use other random sampling methods to draw t in line 169? Like Gaussian? Will different sampling methods affect the algorithm's performance?**
>
> Thanks for the suggestion. $\lambda_t$ is a time-dependent reweighting term (Line 193) that assigns lower weight for noisier $x_t$, which is derived from the KL term in Equation (1) as proved in Appendix B.2.
>
> Yes. During testing, all values of t from [1, T] will be used to denote the denoising phase, so the model should be taught to handle all t values. Other sampling methods beyond uniform sampling are also possible. For example, loss-weighted sampling dynamically adjusts based on the model's learning situation, e.g., by increasing the sample proportion of the currently learned, poorer-performing t, which can lead to a slight improvement but within 1 % action accuracy. Gaussian sampling primarily focuses on sampling more frequently around the mean value and the performance in learning t in low-density regions may not be well. We found Gaussian underperforms uniform sampling by around 3% action accuracy. We've added sampling details in the updated version.
>
> **Q1: In Figures 3(a) and (b), why DiffuSearch only runs around 7 depths? Are there any technical limitations that might prevent running DiffuSearch at greater depths?**
>
> Thanks for mentioning this point. We have already observed a significant improvement from step 1 to step 7, so we have not considered increasing the depth further. We expect that continuing to increase the depth will lead to further performance improvements. However, the main reason is due to resource and time constraints instead of technical limitations. For example, collecting data previously required 1 week using 1024 CPUs (with a depth mostly around 8), and data with a depth of 24 or 25 might take 3 weeks by trebling the search time in Stockfish. Therefore, we are unable to provide a specific number for depth 24 or 25 at this time. However, this is a good suggestion to investigate the convergence of depth in DiffuSearch, and we will update our manuscript once we have results.
>
> **Q2: I noticed that the demonstration plots like Figures 4 and 5 are limited to the comparison between DiffuSearch and Transformer (S-A). Can the authors explain the possible reasons? What about DiffuSearch against Transformer (S-V), against Transformer (SA-V), and Transformer (100 MCTS simulations)?**
>
> Thanks for the question and suggestion. In the latest manuscript, we have also added the results for Transformer S-A, Transformer SA-V, and Transformer with MCTS in Figure 6 to make the paper more complete.
>
> We hope our response could address your questions!

---

> > ### Comment · Reviewer_KzTF · 2024-11-27
> > **Rebuttal**
> >
> > Thank you for the rebuttal and the interesting side project on Lichess. The authors indeed addressed most of my concerns. However, I strongly encourage the authors to increase the search depth in Figure 3 in the final version of this work, as it can help validate their expectation as well: **"We expect that continuing to increase the depth will lead to further performance improvements"**. I need more evidence to be convinced. At the current stage, I will maintain my positive score and increase my confidence.

---

> > > ### Author Response · Authors · 2024-11-27
> > >
> > > Thank you for your feedback about the rebuttal and the side project, and we are pleased to know most of your concerns have been addressed. Regarding further increasing search depth, we believe that the results of DiffuSearch with a depth of 7 already outperforming the one-step policy with explicit search over a depth of 25 have demonstrated the effectiveness of DiffuSearch.
> > > Furthermore, please note that a depth of 7 is already substantial compared to other search-enhanced one-step policy work (e.g., ToT [1]), which is only limited to 3. For your curiosity and to further enhance the model's capabilities, we are conducting experiments on further increasing search depth. However, due to resource constraints, these experiments are still ongoing. We will post the results as soon as they are available.
> > >
> > > [1] Shunyu Yao, Dian Yu, Jeffrey Zhao, Izhak Shafran, Tom Griffiths, Yuan Cao, and Karthik Narasimhan. Tree of thoughts: Deliberate problem solving with large language models. Advances in Neural Information Processing Systems, 36, 2024.

---

### Official Review · Reviewer_SREQ · 2024-11-04

**Soundness:** 2
**Presentation:** 2
**Contribution:** 2
**Rating:** 3
**Confidence:** 5

**Summary:**

The paper proposes to address Chess playing with an implicit search using diffusion modeling. The paper is compared to Transformers networks and MCTS. It reaches an Elo of 1728 when trained on data from Stockfish.

**Strengths:**

The paper compares the proposed approach to other approaches

**Weaknesses:**

The resulting Chess program is very weak compared to the current state of the art. Lc0 or Stoofvless use MCTS and deep neural networks and have a Elo greater than 3500 according to the Swedish Rating List. This is far above the 1728 Elo of DiffuSearch.

**Questions:**

What would you propose to improve the Elo of your program?

---

> ### Author Response · Authors · 2024-11-21
>
> We sincerely thank Reviewer SREQ for the review and are grateful for the time you spent with our submission. We wish to address your confusion and concerns by providing detailed responses to each of your comments.
>
> **W1: The resulting Chess program is very weak compared to the current state of the art. Lc0 or Stoofvless use MCTS and deep neural networks and have a Elo greater than 3500 according to the Swedish Rating List. This is far above the 1728 Elo of DiffuSearch.**
>
> Our research question is to explore whether there are alternative solutions to enhance the planning abilities of LLMs without relying on explicit search techniques like MCTS for solving complex problems. We introduce DiffuSearch based on discrete diffusion as our solution, using chess as our study task, where explicit search is known to be essential. Therefore, the core objective of our work is not to achieve SOTA Elo performance that surpasses all public engines in chess (e.g., lc0 which often relies on a large amount of training data and domain-specific improvements), but rather to use chess as a case study to investigate the above research question through controlled experiments (e.g., data and model size), which can provide insights into building LLMs with enhanced reasoning and planning capabilities.
>
>
> **Q1: What would you propose to improve the Elo of your program?**
>
> In this paper, we frame our discussion within a similar setting to that of [1], where we achieve improvements using fewer resources.
> We propose DiffuSearch, a model that performs implicit search by looking into future states through discrete diffusion modeling, and we have found that effectively modeling the future world within the policy model can enhance the performance (e.g., Elo on chess) without relying on explicit search techniques like MCTS.
>
> The transformer-based one-step policy baselines studied in our work can be also seen as a similar smaller version of that presented in the recent [lc0 blog](https://www.notion.so/2f82d962dbe4495182fe106022576e75?pvs=21), where the lc0 team has find it outperforms original convolution-based models.
> Given the experiments in our work, we anticipate that our approach could further enhance state-of-the-art systems as well, although this is not the primary focus of our research question in this paper.
>
> [1] Ruoss, A., Delétang, G., Medapati, S., Grau-Moya, J., Wenliang, L. K., Catt, E., ... & Genewein, T. (2024). Amortized Planning with Large-Scale Transformers: A Case Study on Chess. NeurlPS'24.

---

> > ### Author Response · Authors · 2024-11-27
> >
> > Dear Reviewer SREQ,
> >
> > Thank you for your valuable time to review our work and for your constructive feedback. As the author-reviewer discussion period is coming to a close, we wonder if you could kindly take a look at both the revision and our response to your comments. We would appreciate it if you could consider adjusting the score based on our responses and the other review comments. If you have any further questions, we are happy to discuss them!
> >
> > Best regards,
> >
> > Authors

---

> > > ### Author Response · Authors · 2024-12-03
> > >
> > > Dear Reviewer SREQ,
> > >
> > > Thank you for your valuable time to review our work and for your constructive feedback. As the author-reviewer discussion period is coming to a close, we wonder if you could kindly take a look at both the revision and our response to your comments. We would appreciate it if you could consider adjusting the score based on our responses and the other review comments. If you have any further questions, we are happy to discuss them!
> > >
> > > Best regards,
> > >
> > > Authors

---

### Author Response · Authors · 2024-11-21
**Summary of updates**

We sincerely thank all reviewers for the time you spent with our submission. We would like to make our main point again first. Our core research question is to explore whether there are alternative solutions to enhance the planning abilities of LLMs without relying on explicit search techniques like MCTS for solving complex problems. We introduce DiffuSearch based on discrete diffusion as our solution, using chess as our study task, where explicit search is known to be essential. We reveal the significant potential of such an implicit search paradigm, which can further provide insights into building LLMs with enhanced reasoning and planning capabilities.

In summary, we made the following updates:

- We have attached code to reproduce the results in the paper.
- While our primary focus isn't on developing a top-performing chess engine, we do host DiffuSearch on Lichess as a side project for anyone interested in playing with it: https://lichess.org/@/diffusearchv0
- We have updated Figure 1 to make both explicit and implicit search clearer, and moved it to introduction.
- We have improved some descriptions based on reviewers’ suggestions (marked in blue in the manuscript).
- We have included a detailed description of the one-step policy with MCTS baseline and training example in Appendix A and C.3, as well as cases predicted from all baselines in Figure 6.

---

### Meta-Review · Area_Chair_SJki · 2024-12-22

**Metareview:**

This paper proposes DiffuSearch, a novel method to enhance the planning abilities of Large Language Models without relying on explicit search methods such as MCTS. By leveraging discrete diffusion modeling, DiffuSearch enables implicit search through future state prediction, demonstrated in the domain of chess. Experiments reveal DiffuSearch outperforms both searchless policies and explicit search-enhanced policies in action accuracy and Elo ratings.

**Strengths**

- DiffuSearch introduces a novel paradigm to planning in LLMs.
- Experiments provide evidence that DiffuSearch outperforms baseline methods.
- DiffuSearch is sample efficient (R4).
- The theoretical framework, including proofs and algorithmic details, is well-organized and sound (R2).

**Weaknesses**

- Broader validation across multiple domains is lacking to generalize the proposed methodology.
- The paper does not fully compare DiffuSearch to recent state-of-the-art transformer-based chess engines (R1, R4)
- Results at greater search depths remain unexplored due to resource limitations (R2, R3).
- Several reviewers noted issues with imprecise terminology and ambiguities in the descriptions of baselines and experimental setups.

While the paper has some limitations, the methodology is innovative, the results are substantial relative to the baselines, and the paper demonstrates potential for future advancements. The rebuttal addressed most concerns adequately.

**Additional Comments On Reviewer Discussion:**

- R2, R3 requested experiments with greater depths to validate DiffuSearch's convergence beyond depth 7. The authors explained that resource constraints prevented deeper evaluations but noted that DiffuSearch showed significant improvements within the tested range. They expressed optimism about future scalability.
- R3 requested more detailed descriptions of baselines. R1, R4 noted the lack of comparison to state-of-the-art chess AI. The authors improved descriptions of the MCTS baseline and included comparisons to additional baselines in revised figures. However, they acknowledged their resource constraints limited comparisons to state-of-the-art chess AI.
- R3, R4 criticized imprecise or misleading claims in the introduction and text. The authors revised imprecise terminology.
- R4 requested a clearer analysis of DiffuSearch's FLOPs during training versus testing. The authors provided additional data on FLOPs.
- Several reviewers flagged an invalid code link. The authors provided the code in the rebuttal.

Overall, the rebuttal significantly improved the paper's clarity and addressed most critical concerns.

---

### Decision · Program_Chairs · 2025-01-22

Accept (Poster)